# Reciprocal virulence and resistance polymorphism in the relationship between *Toxoplasma gondii* and the house mouse

**Jingtao Lilue[1†], Urs Benedikt Müller[1†], Tobias Steinfeldt[1], Jonathan C Howard[1,2]\***

[1]Institute for Genetics, University of Cologne, Cologne, Germany; [2]Fundação Calouste Gulbenkian, Instituto Gulbenkian de Ciência, Oeiras, Portugal

**Abstract** Virulence in the ubiquitous intracellular protozoon *Toxoplasma gondii* for its natural intermediate host, the mouse, appears paradoxical from an evolutionary standpoint because death of the mouse before encystment interrupts the parasite life cycle. Virulent *T. gondii* strains secrete kinases and pseudokinases that inactivate the immunity-related GTPases (IRG proteins) responsible for mouse resistance to avirulent strains. Such considerations stimulated a search for IRG alleles unknown in laboratory mice that might confer resistance to virulent strains of *T. gondii*. We report that the mouse IRG system shows extraordinary polymorphic complexity in the wild. We describe an IRG haplotype from a wild-derived mouse strain that confers resistance against virulent parasites by interference with the virulent kinase complex. In such hosts virulent strains can encyst, hinting at an explanation for the evolution of virulence polymorphism in *T. gondii*.

## Introduction

A virulent parasite that overcomes the immune system and kills its host may seem to have won the confrontation, but it is a Pyrrhic victory when the early death of the host reduces the probability of parasite transmission. Indeed it is in the interests of all hosts and most parasites to prolong the encounter. In this sense, virulence is a failure of co-adaptation. Haldane's conjecture (*Haldane, 1949*) that intense and fluctuating selection imposed by parasites will generate host protein polymorphism is widely accepted (*Woolhouse et al., 2002*; *Clark et al., 2007*; *Kosiol et al., 2008*; *Fumagalli et al., 2011*). The presence in a population of multiple host resistance alleles confronting multiple parasite virulence alleles may reflect a dynamic equilibrium permissive for the persistence of both parties. However this equilibrium is achieved only at the expense of individual interactions fatal for either the parasite or the host, as a consequence of confrontations of inappropriate alleles. In mammals, however, the ability of the adaptive immune system to respond within the time scale of an individual infection and to remember for a lifetime, buffers individuals against dangerous genetic novelty arising from parasites. As a result, life or death outcomes for common infectious diseases in mammals are not generally determined by single, highly penetrant, polymorphic genes. We here report such a case, involving infection of the house mouse, *Mus musculus,* with the ubiquitous intracellular protozoan parasite, *Toxoplasma gondii*. *T. gondii* has a complex life cycle (*Dubey, 1998*) (*Figure 1*). The sexual process occurs in true cats (Felidae) and intermediate hosts become infected by ingesting oocysts spread in cat faeces. A phase of fast intracellular replication and spread (tachyzoite phase) stimulates immunity in the intermediate host, and this in turn induces parasite encystment in brain and muscle cells and lifelong persistence. Predation of the infected host by a cat completes the life cycle. If immunity fails, tachyzoite replication continues uninterrupted, killing the infected host within a few days (*Deckert-Schlüter et al., 1996*). Thus the probability that *T. gondii* completes its life cycle, which is roughly linear with duration of infection of the intermediate host, depends on early immune control.

**\*For correspondence:** jhoward@igc.gulbenkian.pt

†These authors contributed equally to this work

**Competing interests:** The authors declare that no competing interests exist.

**Reviewing editor**: Detlef Weigel, Max Planck Institute for Developmental Biology, Germany

**eLife digest** The parasite *Toxoplasma gondii* is one of the most common parasites worldwide and is known for its unusual life cycle. It reproduces sexually inside its primary host—the cat—and produces eggs that are released in faeces. Other animals, most often rodents, can then become infected when they unknowingly eat the eggs while foraging. Once inside its new host, the parasite reproduces asexually until the rodent's immune system begins to fight back. It then becomes semi-dormant and forms cysts within the brain and muscle cells of its host. In an added twist, the parasite also causes rodents to lose their fear of cats. This increases their chances of being caught and eaten, thereby helping the parasite to return to its primary host and complete its life cycle.

Previous work has shown that virulent strains of *T. gondii* can evade the host immune system in mice by secreting enzymes that inactivate immune-related proteins called IRG proteins. This prevents the infection being cleared and leads to death of the host within a few days. The existence of these virulent strains is intriguing because parasites that kill their host, and thus prevent their own reproduction, should be eliminated from the population. The fact that they are fairly common suggests that there must be a hitherto unknown mechanism that allows rodents to survive these virulent strains.

Lilue et al. now report the existence of such a mechanism in strains of mice found in the wild. In contrast to laboratory mice, wild mice produce IRG proteins that inhibit the enzymes secreted by the virulent strains of *T. gondii*. Moreover, the IRG genes in wild mice are highly variable, whereas laboratory mice all have virtually identical IRG genes.

By uncovering the complexity and variability of IRG genes in wild mice—complexity that has been lost from laboratory strains—Lilue et al. solve the conundrum of how highly virulent *T. gondii* strains can persist in the mouse population, and offer an explanation for the evolution of parasitic strains with differing levels of virulence.

*Mus musculus* is probably the evolutionarily most important intermediate host for *T. gondii*, because it is very abundant worldwide and sympatric with a uniquely abundant felid, the domestic cat. Early immune control of *T. gondii* in mice depends on a family of IFNγ-inducible cytoplasmic effector proteins, the 47 kDa immunity-related GTPases (IRG proteins; for nomenclature of IRG genes and proteins see *Bekpen et al. (2005)*; *Martens and Howard (2006)* and 'Materials and methods') (*Taylor et al., 2000*; *Collazo et al., 2001*; *Liesenfeld et al., 2011*). These assemble on the cytosolic face of the parasito-phorous vacuole membrane (PVM), causing its rupture and killing the included parasite (*Martens et al., 2005*; *Zhao et al., 2009b*). In the C57BL/6 (BL/6) laboratory mouse strain about 20 IRG genes occur in two adjacent clusters on chromosome 11 and one cluster on chromosome 18 (*Bekpen et al., 2005*) (*Figure 2A*). The whole 47 kDa sequences of IRG proteins are typically translated from a single exon. Exceptional are certain 'tandem' IRGB proteins with a molecular weight of about 94 kDa. These genes are transcribed across two chromosomally adjacent IRG coding units and the intergenic spacer is spliced out as an intron resulting in a single open reading frame (*Bekpen et al., 2005*. See also 'Materials and methods' for a note on nomenclature of the tandem genes and proteins). IRG proteins fall into two major functional and sequence sub-families, the GKS group (IRGA, IRGB and IRGD proteins) that are effector proteins at the PVM, and the GMS group (Irgm1, Irgm2 and Irgm3) that are negative regulators of the GKS proteins (*Hunn et al., 2008*). Many strains of *T. gondii* (e.g., the abundant Eurasian strains designated types II and III) are well controlled by the IRG system in laboratory mice, encyst, and are considered avirulent. But others (e.g., type I) are highly virulent (*Sibley and Boothroyd, 1992*), killing the mouse host during the tachyzoite phase of infection. It has very recently been shown that virulence differences between *T. gondii* strains are largely due to polymorphic variation in ROP18 and ROP5 (*Saeij et al., 2006*; *Taylor et al., 2006*; *Khan et al., 2009*; *Behnke et al., 2011*), members of a family of kinases and pseudokinases (*El Hajj et al., 2006*). These proteins are secreted during parasite entry and accumulate on the cytosolic face of the PVM (*Boothroyd and Dubremetz, 2008*). Virulent ROP allotypes inactivate IRG effector proteins by phosphorylating essential threonines in the nucleotide-binding domain (*Fentress et al., 2010*; *Steinfeldt et al., 2010*). In view of the unique importance of *Mus musculus* for the transmission of *T. gondii*, parasite strains acutely lethal for mice should be counterselected. Their presence at a significant frequency

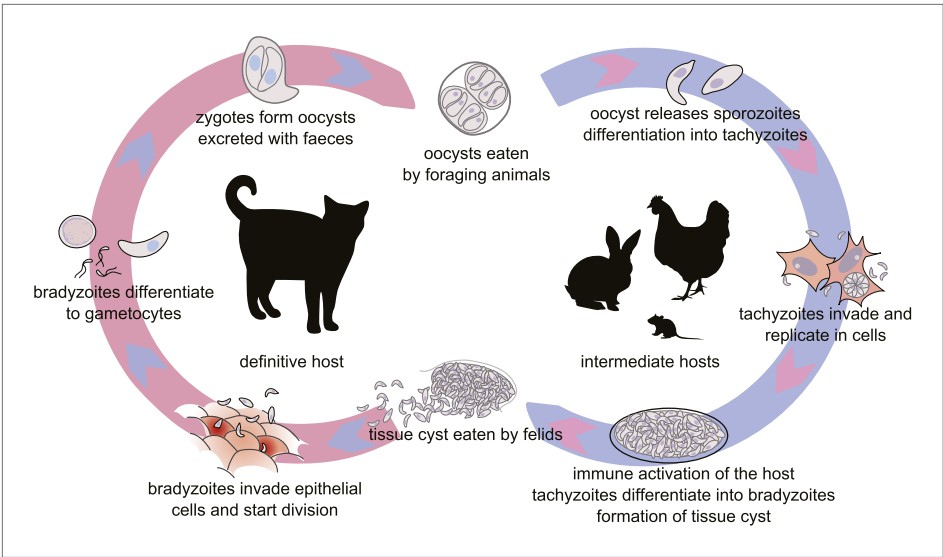

**Figure 1**. The life cycle of *T. gondii*. All warm blooded animals may serve as intermediate hosts, which are infected by ingestion of food or water contaminated by oocysts. Felids, the definitive hosts, are infected by ingesting tissue cysts from their prey. The intermediate phase of the life cycle may be prolonged by carnivory between intermediate hosts (not shown). Modified from free-license pictures.

therefore demands an explanation. In this paper, we reveal that the IRG resistance system in mice has a scale of polymorphic complexity that rivals the MHC, and that a resistant IRG haplotype in the mouse can counter polymorphic virulence factors in the parasite, generating a host phenotype that is permissive for encystment, and thus for the propagation, of virulent strains.

## Results

We compared the IRG genes from a number of mouse strains, mostly those re-sequenced in the Mouse Genomes Project (*Keane et al., 2011*) with the canonical BL/6 sequences (*Figure 2B*). The re-sequenced mice are largely from established laboratory strains. Others are recently derived from the wild and not admixed with laboratory mice. *Mus spretus* is a wild species distinct from *Mus musculus* with a divergent history of 1–3 million years (*Suzuki et al., 2004*). Within laboratory strains we found relatively little IRG protein polymorphism on Chr 11. In contrast, the wild-derived strains showed variation in IRG gene number and remarkable protein polymorphism. For example Irgb6, a protein of 406 residues, had an allele in the CAST/EiJ strain with 47 amino acid substitutions relative to BL/6. On Chr 18, IRG protein sequence variation between laboratory strains was more apparent, and again, wild-derived strains showed extensive sequence polymorphism and copy number variation. The IRG proteins of the outgroup, *M. spretus,* showed considerable protein divergence from *M. musculus* sequences. Pseudogenes occurred in every haplotype and some (e.g. *Irga5* and *Irgb7*) were preserved between *M. musculus* and *M. spretus*. Other IRG sequences were pseudogenes in some haplotypes and apparently intact in others, for example *Irga3* and *Irga8*. Even within this limited group of strains we can distinguish eleven distinct IRG gene haplotypes on Chr 11 and thirteen on Chr 18, already yielding a theoretical population diversity of several hundred IRG genotypes. Diagonal dot-plot comparisons between IRG gene clusters of wild-derived CAST/Ei and MSM/Ms with BL/6 showed IRG genes within tracts of duplication and deletion, associated with numerous repeats in both orientations, a genomic configuration that would be expected to be dynamic even on short time-scales, and doubtless responsible for the two very different dot-plots (*Figure 2C* and *Figure 2—figure supplement 2*).

We added sequences amplified from genomic DNA of wild mice from several Eurasian sites (*Figure 3—figure supplement 1*) to the data from inbred strains to generate nearest neighbour phylograms (*Figure 3A*). We analysed five IRG genes, namely *Irgm1*, a regulatory IRG protein of the GMS class (*Hunn et al., 2008*), and *Irga6*, *Irgb2*, *Irgb6*, and *Irgb10*, all effector GKS proteins localizing

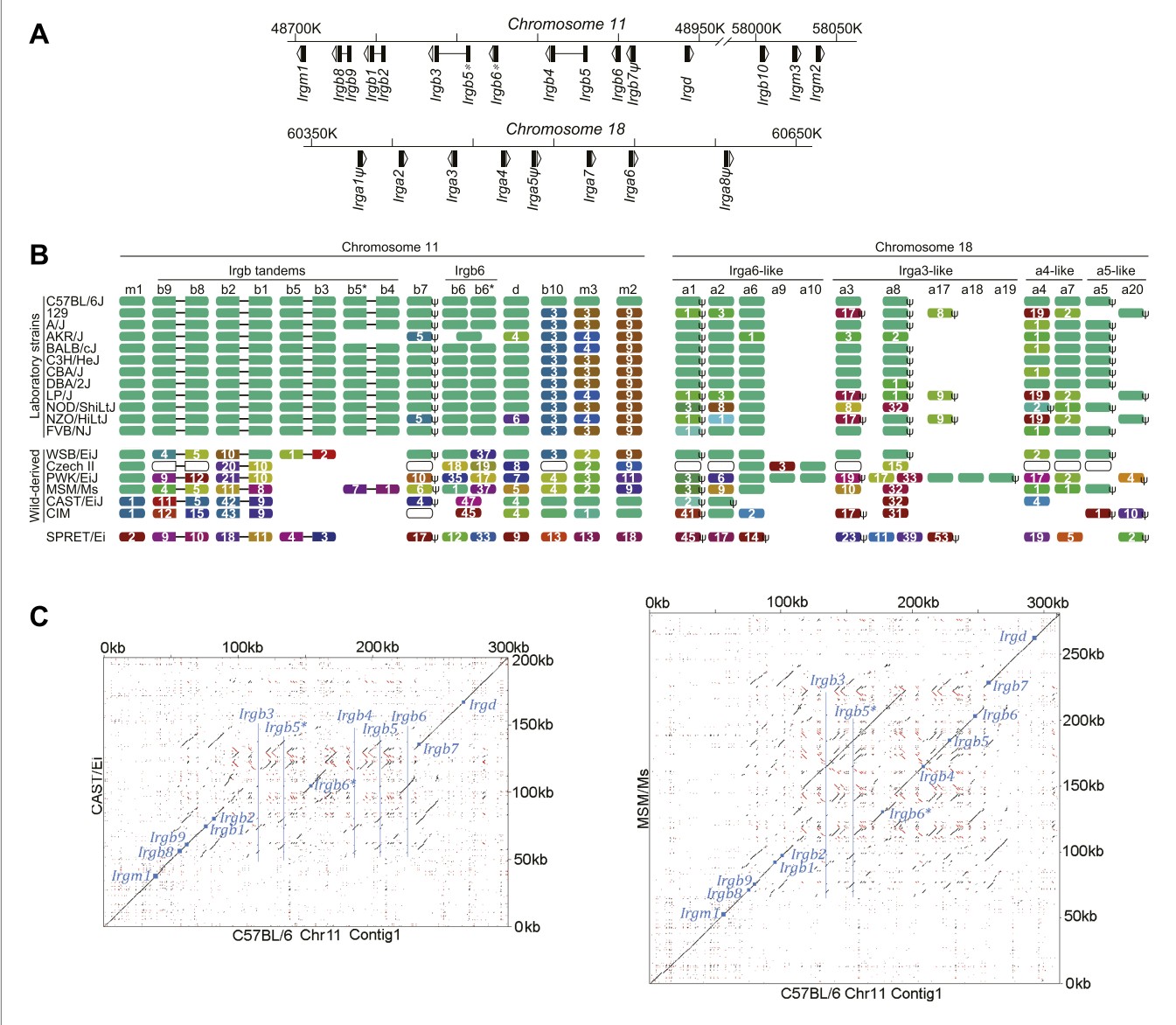

**Figure 2**. IRG protein polymorphism in inbred mouse strains. (**A**) Linear order of IRG gene clusters on Chr 11 and Chr 18 of mouse strain BL/6. (**B**) Polymorphism at the protein level in the IRG genes of Chr 11 and Chr 18. *Irga9–Irga20* are absent from BL/6 and are inferred from resequencing data. Each colour block represents one IRG open reading frame and shows the number of amino acid substitutions/indels relative to the BL/6 allele. The colours of the blocks indicate their phylogenetic relationship (***Figure 2—figure supplement 1***). Open blocks in strains Czech II and CIM indicate homologues expected but not yet found. 'ψ' indicates probable pseudogenes. (**C**) Dot plots of the longer IRG gene clusters on Chr 11 in CAST/Ei and MSM/Ms against the BL/6 genomic sequence. Small blue squares show the positions of homologous coding units, blue lines indicate the positions of genes in BL/6 that are absent from the other genomes.

The following figure supplements are available for figure 2:

**Figure supplement 1**. Unrooted phylogenetic trees of the indicated IRG genes were overlapped with RGB colour wheels.

**Figure supplement 2**. Chr 11 (shorter contig containing *Irgb10*, *Irgm2* and *Irgm3*) of CAST/Ei and MSM/Ms against the BL/6 genomic sequence (Top).

to the *T. gondii* PVM during infection (***Khaminets et al., 2010***). *Irgb2* forms the N-terminal half of the tandem protein, Irgb2-b1. The nearest-neighbour phylograms of *Irgm1*, *Irgb10* and *Irga6* are shallow and the *M. spretus* sequences fall into outgroups. Thus most of the sparse polymorphic variation in these sequences has been acquired since the divergence of *M. musculus* from *M. spretus.* This conclusion

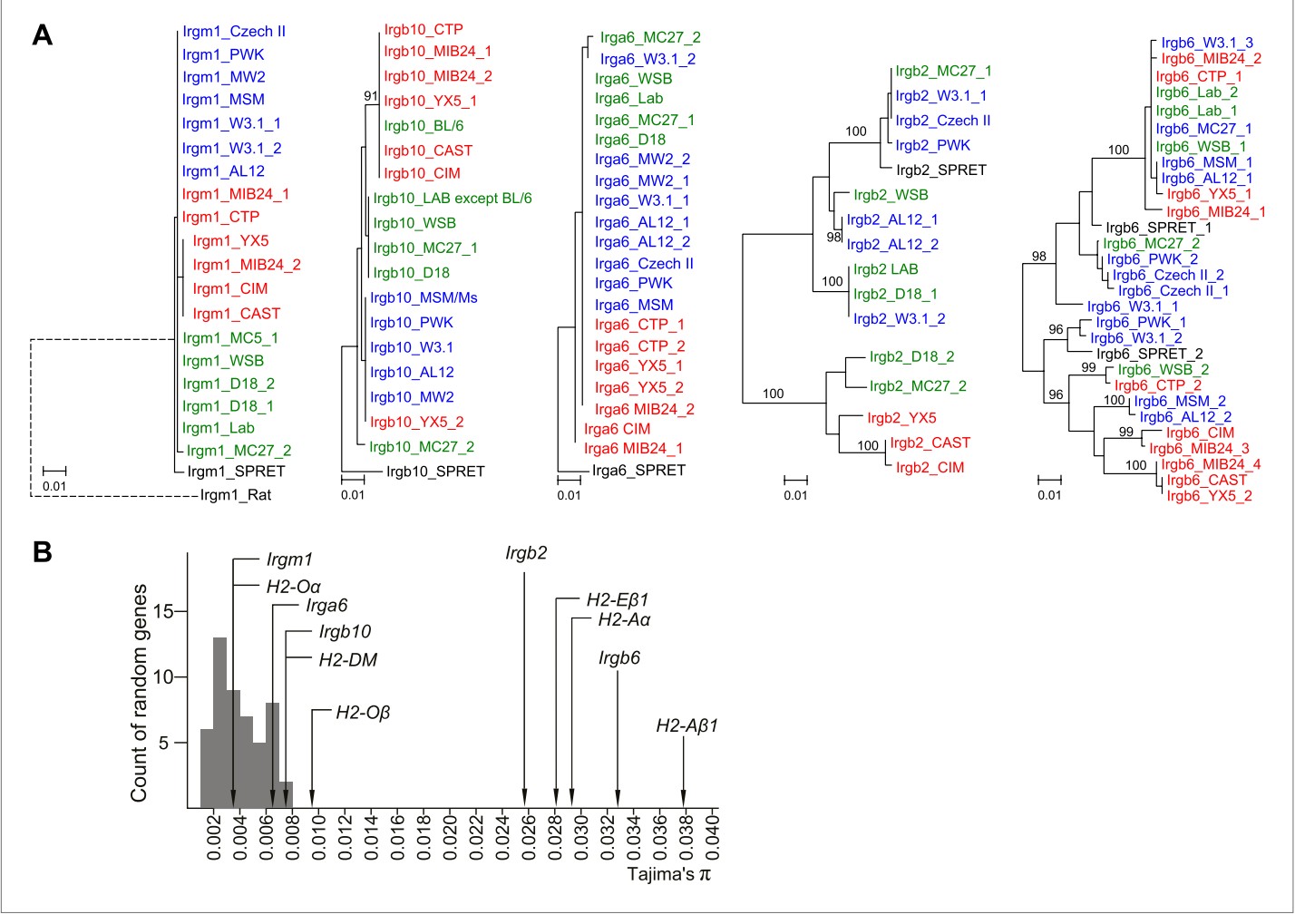

**Figure 3**. Polymorphism of five IRG genes. (**A**) Phylogenetic trees of five IRG genes sequenced from DNA of wild mice collected from various sites in Eurasia. Green, blue and red taxa represent *M. m. domesticus, M. m. musculus* and *M. m. castaneus* samples respectively. The black taxon represents *Mus spretus*. Alleles found in heterozygous condition in certain mice are indicated by numbers appended to individual mouse identifiers (some haplotypes contain 2 *Irgb6* paralogous genes (**Figure 2B**), hence potentially up to 4 alleles). Bootstrap values are shown if >90. The sequences are avaliable in (**Figure 3—source data 2-6**). (**B**) The nucleotide pairwise diversities (π) of genes across seven laboratory and wild-derived inbred mouse strains (BL/6, AKR/J, MSM/Ms, CAST/Ei, PWK/PhJ, WSB/Ei and Spretus/EiJ). Grey bars indicate the distribution of π from 50 'random' genes (**Figure 3—source data 1**). The π values of individual IRG and *MHC* genes are indicated by arrows.

The following source data and figure supplements are available for figure 3:

**Source data 1**. Nucleotide diversities of 50 random genes in seven mouse strains.
**Source data 2**. Alignment of Irgm1 alleles, in FASTA format.
**Source data 3**. Alignment of Irga6 alleles, in FASTA format.
**Source data 4**. Alignment of Irgb2 alleles, in FASTA format.
**Source data 5**. Alignment of Irgb6 alleles, in FASTA format.
**Source data 6**. Alignment of Irgb10 alleles, in FASTA format.
**Figure supplement 1**. Mouse samples collected for this study.
*Figure 3. Continued on next page*

*Figure 3. Continued*

**Figure supplement 2**. Phylogenetic trees of *Irga6, Irgm1* and *Irgb10* in mouse strains and wild mice.

**Figure supplement 3**. Phylogenetic tree of *Irgb2* in mouse strains and wild mice.

**Figure supplement 4**. Phylogenetic tree *Irgb6* in mouse strains and wild mice.

is supported by a tendency for individual minor variants to be concentrated in one or other of the three recent, geographically separated and partially isolated subspecies, *Mus musculus musculus*, *Mus musculus domesticus* and *Mus musculus castaneus* (***Figure 3A***, ***Figure 3—figure supplements 2–4***). In contrast, the phylograms for *Irgb2* and *Irgb6* have a depth similar to the mouse-rat divergence (about 20 million years) (***Gibbs et al., 2004***), the *M. spretus* sequences are embedded in the *M. musculus* trees, and any tendency to correlation with subspecies is seen only in the outermost branches. Thus the polymorphism in these two genes is ancient and has persisted through a number of speciation events. The scale of polymorphism in the IRG system estimated by Tajima's $\pi$ from 7 mouse strains resembles that of classical *MHC* genes (***Figure 3B***). That there is also considerable local polymorphism within geographic ranges was confirmed by the identification of numerous heterozygotes at all loci examined by PCR among the wild mouse captures.

With such striking polymorphic diversity in the sequences of proteins known to be involved in resistance against *T. gondii*, it was appropriate to assay the ability of different IRG genotypes to resist virulent *T. gondii*. The wild-derived Indian strain CIM, which has a number of divergent IRG alleles (***Figure 2B***), proved to be remarkably resistant to the type I virulent *T. gondii* strain, GT-1 (***Figure 4A***). All CIM mice survived intraperitoneal infection with GT-1 tachyzoites while all laboratory mice (NMRI strain) died within 15 days. In vitro, tachyzoite proliferation of avirulent strains in mouse cells is inhibited by induction of IRG proteins with IFNγ (***Könen-Waisman and Howard, 2007***). In contrast, proliferation of type I virulent strains is not inhibited (***Zhao et al., 2009c***). By this assay, IFNγ-induced CIM diaphragm-derived cells (DDC see 'Materials and methods') inhibited proliferation of type I virulent RH-YFP strain tachyzoites as efficiently as they inhibited the proliferation of avirulent strains, while BL/6 DDC inhibited only the avirulent strains (***Figure 4B***). Cells from two other *M. m. castaneus* strains, CAST/EiJ and CTP were almost as resistant as CIM. Additionally, IFNγ-induced CIM cells died by reactive cell death after infection with both virulent RH-YFP and avirulent ME49, while BL/6 cells died only after infection with avirulent strains (***Figure 4C***). Reactive death of *T. gondii*-infected mouse cells after induction with IFNγ is associated with IRG protein-mediated host resistance, as previously reported (***Zhao et al., 2009b***).

Resistance of CIM mice to the type I virulent RH-YFP strain in vivo was exploited to test linkage of resistance to the IRG system in a (CIM×BL/6)F$_1$×BL/6 backcross. For typing purposes the IRG gene clusters from CIM and BL/6 were differentiated by RFLPs in *Irga1* (Chr 18) and *Irgb6* (Chr 11). Fluorescent-tagged tachyzoites (RH-YFP) were injected intraperitoneally into mice and the frequency of infected peritoneal cells measured by FACS after 5 days. In five independent experiments involving a total of 65 typed animals (***Figure 5A***), resistance by this assay was almost completely dominant and was tightly linked to the IRG$_{CIM}$ gene cluster on Chr 11 (p<<10$^{-6}$). There was no detectable association of virulence with the IRG$_{CIM}$ cluster on Chr 18. In a similar analysis of 53 F$_2$ progeny, mice typed as homozygous IRG$_{CIM}$ at the Chr 11 gene cluster were as resistant as wild-type CIM mice. Thus the reduced resistance of some mice heterozygous for IRG$_{CIM}$ on Chr 11 may be due to a gene dosage effect at the IRG locus. Assayed directly by survival, 51 backcross and F$_2$ mice infected with RH-YFP followed the same pattern as in the peritoneal cell assay (***Figure 5B***), with the Chr 11 IRG$_{CIM}$ homozygotes (*cc* in ***Figure 5B***) showing complete resistance, the heterozygotes (*bc*) substantial but incomplete resistance and the IRG$_{BL/6}$ homozygotes (*bb*) complete and acute susceptibility.

Recent results have implicated the inflammasome core component Nlrp1 (NLR family, pyrin domain containing 1) in resistance to *T. gondii* in human (***Witola et al., 2011***) and rat (***Sergent et al., 2005***; ***Cavailles et al., 2006***). Since the Nlrp1 complex locus is about 20 Mb telomeric to the IRG system on Chr 11 it was possible that polymorphism at this locus (***Boyden and Dietrich, 2006***) was responsible for the differential resistance apparently linked to the IRG complex. We therefore typed all the backcross progeny shown in ***Figure 5A*** by PCR designed to distinguish the BL/6 and CIM allotypes of the *Nlrp1b* locus (see 'Materials and methods'). We obtained 17% recombinants between *Irgb6* and *Nlrp1*. There

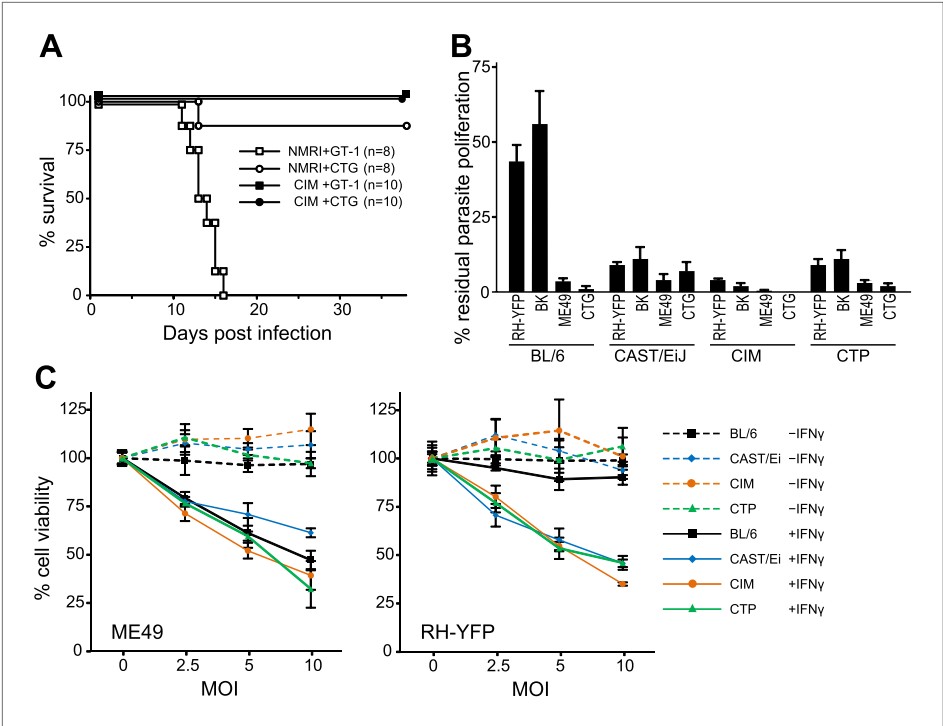

**Figure 4**. Resistance of wild-derived mouse strains to virulent *T. gondii*. (**A**) Cumulative mortality of NMRI and CIM mice infected with 100 or 300 (data pooled) tachyzoites of the indicated *T. gondii* strains. (**B**) IFNγ-mediated growth inhibition of virulent (type I RH-YFP, BK) and avirulent (type II ME49, type III CTG) *T. gondii* strains in DDC of laboratory (BL/6) and wild-derived, inbred mice (CAST/Ei, CIM, CTP). Proliferation of parasites was measured by $^3$H-uracil incorporation and is displayed as percentage of residual *T. gondii* proliferation, as described in 'Materials and methods'. Error bars show standard deviations of quadruplicate values. (**C**) IFNγ-dependent reactive cell death of mouse DDC cell lines infected with *T. gondii*. DDC were either stimulated with 100 U/ml of IFNγ 24 hr prior to infection or left unstimulated. Cells were infected with type II strain ME49 or type I strain RH-YFP at the indicated MOIs for 8 hr. Cell viabilities were measured as described in 'Materials and methods' and expressed as percentages of those recorded for uninfected cells (MOI = 0). Error bars show standard deviations of quadruplicate values.

was no correlation between *Nlrp1* genotype and the ability to clear virulent *T. gondii* (**Figure 5A**). It is anyway unlikely that Nlrp1 plays a role in the resistance polymorphism we describe since it is not detectably expressed in the IFNγ-induced DDC transcriptomes (data not shown).

The existence of mouse genotypes resistant to virulent *T. gondii* strains is consistent with a co-evolutionary explanation for the evolution of virulence. To sustain the argument, however, it would be necessary to show that *T. gondii* strains that are lethal in laboratory mice, and thereby suffer a major cost, can form functional cysts in resistant CIM mice, permitting their propagation. We therefore searched for brain cysts in CIM mice infected 6–8 weeks earlier with virulent type I or avirulent type III strains of *T. gondii*. Two type I virulent strains, GT-1 and BK, formed cysts in CIM mice (**Figure 5C**, **Table 1**). Thus the resistance of the CIM mouse provides an adaptive niche for highly virulent *T. gondii* strains. The avirulent type III strain, NED, encysted in the laboratory mice >20× more efficiently than in CIM mice (**Table 1**).

The IRG resistance mechanism operates cell-autonomously, and in BL/6 cells the loading of IRG proteins onto the parasitophorous vacuole occurs only with avirulent strains even in cells infected simultaneously with virulent and avirulent strains (**Zhao et al., 2009a**; **Khaminets et al., 2010**). Consistently, the resistance of the CIM strain against virulent *T. gondii* was reflected in the behaviour of IRG proteins in IFNγ-induced CIM-derived cells infected with virulent RH-YFP strain *T. gondii*. In BL/6 cells Irgb6$_{BL/6}$ loaded onto only about 8% of vacuoles, while in CIM cells the highly divergent Irgb6$_{CIM}$ protein loaded onto more than 50% of vacuoles (**Figure 6A**). Even more extreme, the tandem protein, Irgb2-b1$_{BL/6,}$ which is poorly expressed in BL/6 cells (**Figure 6B**), was not detectable on the RH-YFP PVM, while the highly divergent Irgb2-b1$_{CIM}$ was well-expressed in CIM cells and loaded onto over 95% of vacuoles (**Figure 6A**).

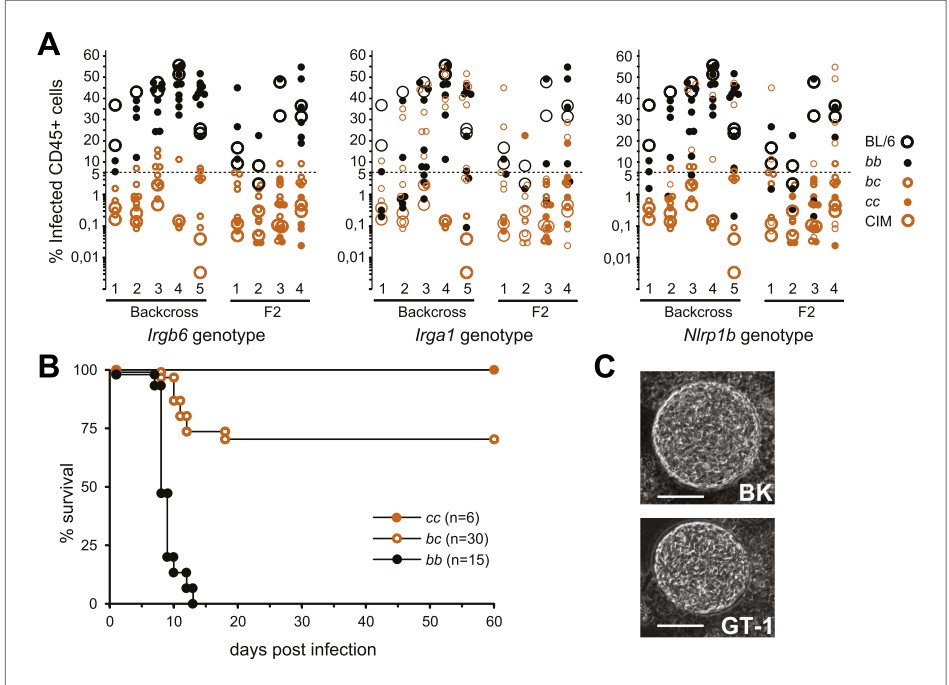

**Figure 5**. Resistance of CIM mice to virulent *T. gondii* is dependent on the Chr 11 IRG locus. (**A**) Infected CD45+ peritoneal cells in (BL/6×CIM)F$_1$×BL/6 backcross (5 experiments, total 83 mice) and (BL/6×CIM)F$_2$ mice (4 experiments, total 69 mice) 5 days after i.p. injection of 500 RH-YFP tachyzoites. Genotypes at IRG loci and at the Nalp1 locus for backcross and F$_2$ mice are shown (see key) as *bb* (homozygous BL/6), *bc* (heterozygous BL/6/CIM) or *cc* (homozygous CIM). Elimination of infected cells is linked to the CIM haplotype on Chr 11 (n.b. the y-axis is logarithmic below 5%). (**B**) Cumulative mortality of (BL/6×CIM)F$_1$×BL/6 and (BL/6×CIM)F$_2$ mice infected with 500 RH-YFP tachyzoites. *Irgb6* genotypes are shown as in (**A**). (**C**) Cysts of type I *T. gondii* strains in brain homogenates of CIM mice infected 6–8 weeks earlier (quantitation in ***Table 1***). Bar = 20 μm.

The loading of the effector IRG protein Irga6 onto the parasitophorous vacuole of virulent *T. gondii* strains is prevented by a parasite-derived kinase complex (ROP5/ROP18) that phosphorylates two threonines that are essential for IRG protein function (***Steinfeldt et al., 2010***). Remarkably, in CIM cells infected with RH-YFP, phosphorylated Irga6$_{CIM}$ was barely detectable with an antiserum specific for Irga6$_{BL/6}$ phosphorylated at T108 (***Figure 6C***). Irga6$_{CIM}$ differs from Irga6$_{BL/6}$ at only two residues, both distant from the phosphorylation sites and was phosphorylated normally when transfected into IFNγ-induced L929 cells infected with virulent RH-YFP (***Figure 6D***). Thus Irga6$_{CIM}$ apparently remains unphosphorylated in CIM cells as a result of active inhibition of the parasite kinase complex. The following experiment showed that the highly polymorphic tandem protein, Irgb2-b1$_{CIM}$, is largely responsible.

**Table 1.** Cyst counts in *T. gondii* infected mice

| Mouse | UIC* | CIM | CIM | CIM | CIM | CIM | CIM | CIM | NMRI | NMRI | NMRI | NMRI |
|---|---|---|---|---|---|---|---|---|---|---|---|---|
| *T. gondii* (# injected) | – | GT-1 500 | GT-1 1000 | BK 5000 | BK 10,000 | NED 10,000 | NED 10,000 | NED 10,000 | GT-1 500 | BK 500 | NED 10,000 | NED 10,000 |
| Q-PCR (cycle) | >35 | 22.9 | 24.2 | 18.7 | 22.0 | 26.9 | 26.2 | 28.8 | Dead | Dead | 22.6 | 20.3 |
| Cysts per brain | 0 | 100 | 50 | 130 | 150 | 220 | 90 | 15 | | | 720 | 4800 |
| Antibody test | – | + | + | + | + | + | + | + | | | + | + |

*Uninfected control.

Mice were sacrificed 5 weeks (NED) or 6–8 weeks (BK and GT-1) after tachyzoite injection. Infection was verified by serum antibody. Cysts were evaluated by direct counting in homogenised brains and by quantitative PCR of a repeat element of *T. gondii* (***Reischl et al., 2003***) in genomic DNA samples isolated from mouse brains.

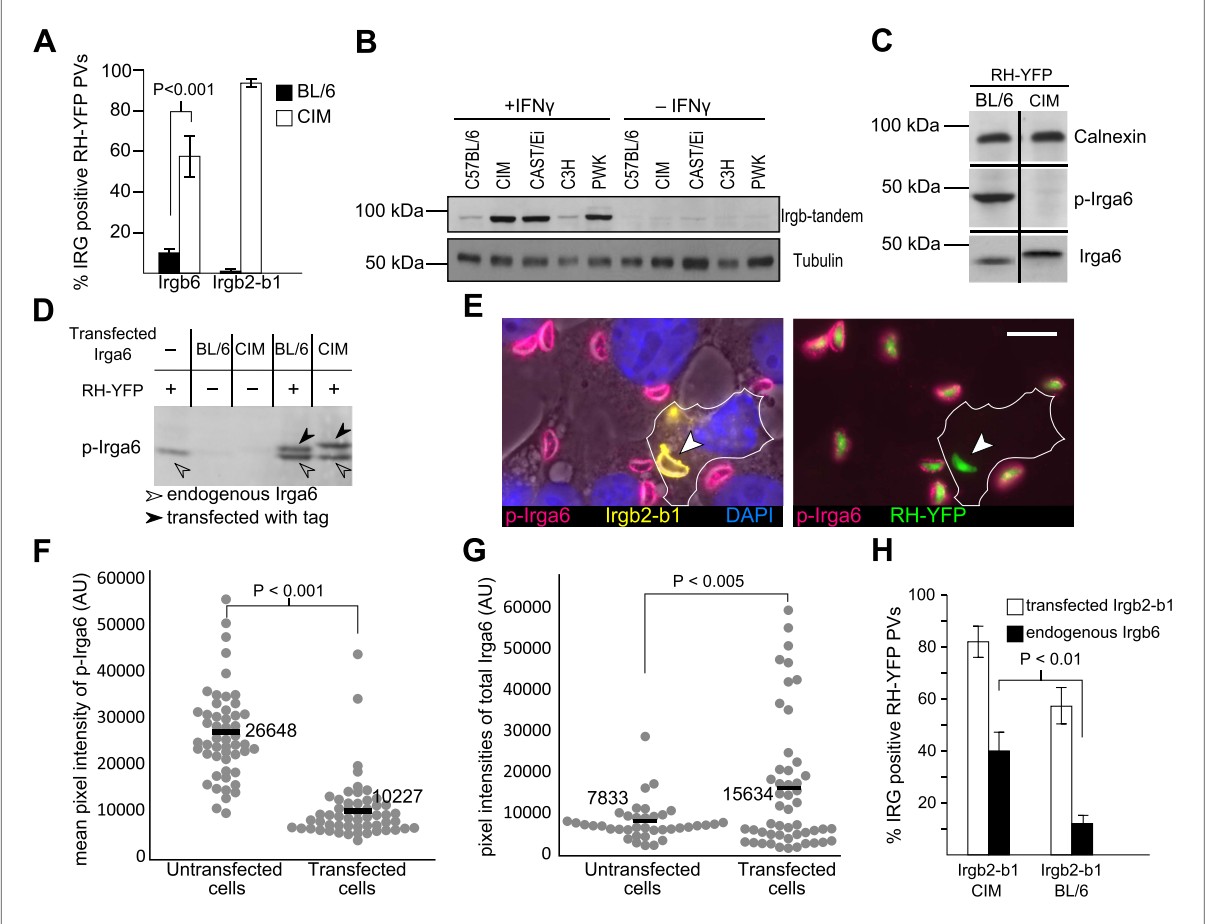

**Figure 6**. Irgb2-b1$_{CIM}$ protects other IRG members from inactivation. (**A**) Immunofluorescent quantitation of loading of Irgb6 and Irgb2-b1 on to RH-YFP vacuoles in IFNγ-induced BL/6 and CIM DDC. (**B**) Strain dependence of expression levels of Irgb-tandem proteins. (**C**) Reduced phosphorylation of Irga6$_{CIM}$ on T108 by RH-YFP in IFNγ-induced CIM DDC (chopped western blot for calnexin, phosphorylated Irga6 and total Irga6). n.b. Irga6$_{CIM}$ characteristically runs at a higher apparent molecular weight than Irga6$_{BL/6}$ (**D**) Transfected Irga6$_{CIM}$ and Irga6$_{BL/6}$ are both phosphorylated in IFNγ-induced L929 cells infected with RH-YFP as shown in western blot of detergent lysates. Phosphorylation of Irga6 is indicated by a size-shift (black arrowhead) for both Irga6$_{BL/6}$ and Irga6$_{CIM}$ in infected cells. The lower band in the two transfected/infected tracks is the endogenous Irga6$_{BL/6}$. (**E**) Irgb2-b1$_{CIM}$ (yellow) transfected into IFNγ-induced BL/6 MEFs inhibits phosphorylation of Irga6 (red) by RH-YFP seen in untransfected cells serving as control. Bar = 10 µm. (**F**) Immunofluorescent quantitation of phosphorylated Irga6 on the PVM of RH-YFP in Irgb2-b1$_{CIM}$-transfected cells and untransfected cells prepared in (**E**). (**G**) Immunofluorescent quantitation of total Irga6 on the PVM of RH-YFP in Irgb2-b1$_{CIM}$ transfected and untransfected cells prepared in (**E**). (**H**) Enumeration of Irgb6-positive vacuoles in BL/6 MEFs induced by IFNγ and transfected with Irgb2-b1$_{CIM}$ or Irgb2-b1$_{BL/6}$.

Irgb2-b1$_{CIM}$ was transfected into IFNγ-induced BL/6 mouse embryonic fibroblasts (MEFs) infected with RH-YFP virulent strain *T. gondii*. Phosphorylated Irga6 was measured at the PVs in transfected cells expressing Irgb2-b1$_{CIM}$ and in untransfected cells. *Figure 6E,F* show that the amount of phosphorylated Irga6 was strikingly reduced on vacuoles loaded with Irgb2-b1$_{CIM}$ while the amount of total Irga6 loaded onto Irgb2-b1$_{CIM}$ positive vacuoles was increased (*Figure 6G*). Thus the decreased signal of phosphorylated Irga6 on the PVM was not caused by competition for loading between Irga6 and Irgb2-b1, but rather by inhibition of phosphorylation. Transfected Irgb2-b1$_{CIM}$ also stimulated the loading of endogenous Irgb6$_{BL/6}$ onto RH-YFP vacuoles (*Figure 6H*). Transfected Irgb2-b1$_{BL/6}$, which is well expressed unlike the endogenous protein, had little or no effect on the loading of Irgb6$_{BL/6}$ (*Figure 6H*). Thus the essential difference between Irgb2-b1$_{BL/6}$ and Irgb2-b1$_{CIM}$ in determining resistance lies in the amino acid sequence polymorphism rather than in the protein expression level.

Preliminary results suggest that Irgb2-b1$_{CIM}$ may bind directly to the protein product of the virulent allele of ROP5. Thus loading of Irgb2-b1$_{CIM}$ onto the PVM of ROP5-deficient RH strain parasites in CIM cells was greatly reduced, and consistently, the amount of loading on to different RH-related strains

correlated with strain-specific variation in the amount of ROP5 (*Figure 7A*). Irgb2-b1$_{CIM}$ itself becomes phosphorylated during infection with virulent *T. gondii* (*Figure 7B*), thus is also a target for the active ROP5-ROP18 kinase complex, suggesting that Irgb2-b1$_{CIM}$ may block phosphorylation of Irga6 by binding ROP5 pseudokinase at the vacuole, thereby distracting rather than inhibiting ROP18 kinase. ROP5 binds to Irga6 via helix 4 (H4) of the Irga6 nucleotide binding domain (*Fleckenstein et al., 2012*), so a homologous structure on Irgb2-b1 may be involved in ROP5 interaction. Indeed the putative H4 and αD structural domains of the Irgb2 subunit are highly polymorphic and show recent divergent selection (*Figure 7C*), indicating possible co-evolution with ROP kinases and pseudokinases. The high polymorphism of H4 in Irgb2-b1 is also consistent with a direct interaction with a polymorphic component of the pathogen. If Irgb2-b1 interacts with a host protein to bridge to ROP5 the interaction surface between the two host proteins would not be expected to evolve rapidly under divergent selection.

## Discussion

We have shown that the IRG protein system essential for resistance against *T. gondii* infection in the mouse has a complex polymorphism on the scale of the MHC, and that at least one IRG haplotype, found in the wild-derived CIM strain mouse, is strikingly resistant to *T. gondii* strains that are highly virulent for laboratory mice. We also provide a mechanistic explanation for the resistance of the CIM

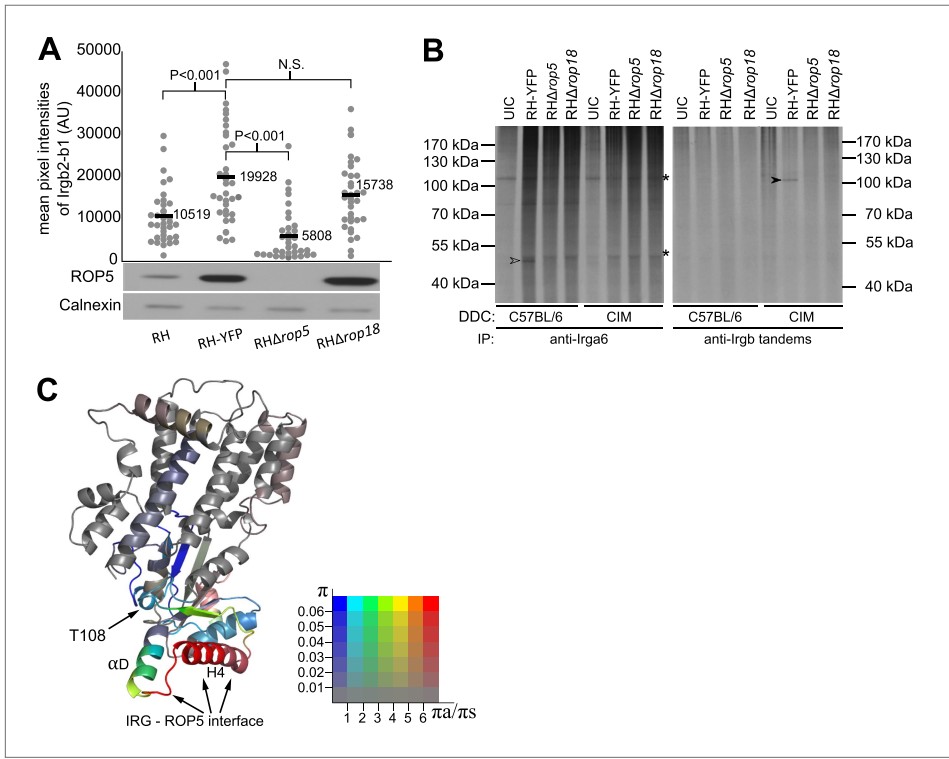

**Figure 7**. IRG-tandem proteins interact with virulence factors. (**A**) Loading of Irgb2-b1$_{CIM}$ on to vacuoles of RH variants in IFNγ-induced CIM DDC. Irgb2-b1$_{CIM}$ loading (dot plots, upper image) is positively correlated with ROP5 expression level in the parasite (western blot, lower image). (**B**) Autoradiogram ($^{33}$P) of immunoprecipitated IRG proteins in DDC infected with *T. gondii*. Irga6 was phosphorylated (open arrow head) by virulence factors of *T. gondii* in BL/6 DDC, but not in CIM DDC. Irgb-tandem proteins were phosphorylated only in CIM DDC (filled arrow head) in a ROP5/ROP18-dependent manner. UIC = uninfected control. Asterisk (*) indicates non-specific phosphorylated proteins (**C**) Ribbon model of Irgb2 predicted based on the structure of Irga6, showing diversifying selection associated with H4 and αD of the G-domain. The colours of the ribbons are based on the π and πa/πs values among 40 alleles sequenced from mouse strains and wild mice with a 120 bp slide window and 10 bp step. The colours indicate the πa/πs value, from purifying selection (blue) to significantly diversifying selection (red). The saturations of colours are defined by overall π value in the sliding window, indicate conserved regions of the protein (low saturation) to highly polymorphic regions (high saturation).

mouse against type I virulent *T. gondii* strains. Resistance is determined by the presence of the polymorphic tandem IRG protein, Irgb2-b1$_{CIM}$ encoded on Chr 11, which blocks the ROP5/ROP18 kinase complex of the virulent parasite, preventing phosphorylation and consequent inactivation of IRG effector proteins. Taken together, our results suggest a selective explanation for the evolution of *T. gondii* strains that are highly virulent for certain mice. If the mouse is an evolutionarily significant host for *T. gondii* the parasite must balance its virulence against mouse resistance, in order to allow encystment. To minimise the cost of infection, selection on the mouse favours the evolution of strong resistance alleles. This in turn leads to the selection of parasite strains able to counteract heightened resistance sufficiently to allow encystment in these mouse genotypes, as type I *T. gondii* strains in CIM mice. However such virulent *T. gondii* strains are counterselected by acute lethality in less resistant mice, while less resistant mice are better hosts for less virulent *T. gondii* strains.

Why, however, are not all mice highly resistant? Loss of the entire IRG system in several vertebrate groups, for example higher primates and birds (*Bekpen et al., 2005*), suggests that possession of the IRG system may be costly. Highly resistant genotypes may be more costly than less resistant ones. Alternatively, molecular specificity in interactions between polymorphic parasite virulence factors and mouse IRG proteins may favour IRG genotypes that are highly resistant to some *T. gondii* genotypes but more susceptible to others. Such an allele-specific system is familiar from polymorphic plant disease resistance (*R*) genes and strain-specific pathogen virulence and avirulence (*Flor 1971*; *Dodds et al., 2006*), where exact molecular matches or mismatches determine the outcome of a given infection. These alternative models can now be analysed formally and tested experimentally.

Much of the argument in this paper has focused on the importance of the house mouse as an intermediate host and vector for *T. gondii*, resting on the global abundance of the species and its sympatry with the cat, and obviously supported by the intimate antagonistic relationship between *T. gondii* virulence factors and mouse resistance factors. However, although many bacteria and protozoal parasites are not resisted by the IRG system, *T. gondii* is not the only organism that is. At least two members of the *Chlamydia* species complex are resisted by the IRG system in mice, and polymorphic variation on Chr 11, possibly associated with Irgb10, affects the level of resistance (*Bernstein-Hanley et al., 2006*; *Miyairi et al., 2007*). There will surely be other organisms that may contribute to the genomic complexity and polymorphism of the IRG system. For example, the massive and ancient polymorphism of Irgb6 is not directly accounted for by our experiments. In in vitro experiments in BL/6 cells transfected Irgb2-b1$_{CIM}$ protects Irga6 from phosphorylation in the absence of the CIM allotypes of the other IRG proteins including Irgb6 (*Figure 6*) and in the CAST/Ei strain, which is almost as resistant as CIM against *T. gondii*, Irgb6 is barely expressed (unpublished results). It may be that Irgb6 polymorphism reflects a pattern of resistance against another abundant mouse pathogen. However it is also not excluded that Irgb6 polymorphisms relate to further virulence polymorphism in *T. gondii* not expressed in the limited range of strains used in the present study. Likewise, mice are not the only intermediate hosts for *T. gondii*. The parasite infects all known mammals and birds, and many other species than *M. musculus* are prey for domestic cats. Furthermore, individuals of other species, for example rats (*Jacobs and Jones, 1950*) and American deer mice (*Peromyscus*) (*Frenkel, 1953*) have been shown to resist infection with strains virulent for laboratory mice. Thus some of the genomic complexity of the ROP system and other virulence-associated *T. gondii* secretory proteins may be relevant to immunity modification in other host species. Nevertheless, despite these distractions, mice and *T. gondii* have clearly had a large selective impact on each other.

The close association of cat and mouse with humans has led to large-scale transformations in the ecology of the parasite over the last 10,000 years. The impact of the dramatic narrowing of the parasite's effective host range on the genetics of the virulence-resistance relationship may help to understand better the co-evolutionary forces at work in this system. The patterns of genetic resistance against *T. gondii* in its most important intermediate hosts will determine the relative abundance of strains contaminating the environment, and thereby the strains available for human infection.

## Materials and methods

### Nomenclature of IRG genes and proteins

The systematic, phylogenetically-based nomenclature of IRG genes and proteins used in this paper was published in *Bekpen et al. (2005)* and replaced the non-systematic naming of some of the

first-discovered members of the family. A partial synonymy relating the new nomenclature to other names that have appeared in the literature is given in *Martens and Howard (2006)*. The root name for members of the IRG family is IRG. This may followed by single letters designating phylogenetically distinct sequence sub-families, as IRGA, IRGB, IRGC, IRGD, IRGM. Individual gene or protein names are given in the standard mouse nomenclatural form, as Irga6 (protein) and *Irga6* (gene) or Irgb6 (protein) and *Irgb6* (gene). A subset of IRGB genes consist of two adjacent IRG coding units that are transcribed together from single promoters and spliced to form 'tandem' IRG proteins with a molecular weight twice that of the individual IRG coding units. In the original study it was not clear whether the individual coding units could also be expressed separately and they were therefore given separate names (as Irgb1 and Irgb2) (*Bekpen et al., 2005*). It is now clear that these coding units are expressed only as tandems. In the present study we have therefore named the tandem genes and proteins with the names of their constituent coding units in N-C order, thus Irgb2-b1 is a tandem IRG protein with the Irgb2 coding unit N-terminal to the Irgb1 coding unit.

## Sequencing and assembly of IRG genes of inbred mouse strains and wild mice

The *Mus musculus* genomes sequenced by the Mouse Genomes Project (MGP) (*Keane et al., 2011*), including 129P2/OlaHsd, 129S1/SvImJ, 129S5/SvEvBrd, A/J, AKR/J, BALB/cJ, C3H/HeJ, C57BL/6NJ, CAST/EiJ, CBA/J, DBA/2J, FVB/N, LP/J, NOD/ShiLtJ, NZO/HlLtJ, PWK/PhJ, WSB/EiJ and the *Mus spretus* genome, SPRET/EiJ, were accessed by LookSeq (http://www.sanger.ac.uk/resources/mouse/genomes/), and the sequences of IRG genes were assembled (for details see below, 'Bioinformatics'). Three 129 strains (129P2/OlaHsd, 129S1/SvImJ and 129S5/SvEvBrd) were identical in all IRG coding units, so they were combined into one virtual strain, 129. Mouse strain C57BL/6NJ was confirmed identical for all IRG genes with C57BL/6J, so the results were combined into one strain, C57BL/6. The sequences of IRG genes of mouse strains Czech II, NMRI and JYG were acquired from the NCBI database and are listed in *Table 2*. The CAST/Ei BAC library CHORI-26 was screened with 40bp synthetic probes for *Irga1, Irga3, Irga4, Irga6, Irgm1, Irgm2, Irgb1, Irgb6, Irgb8* and *Irgb10*. Eight out of 61 positive BAC clones were sent to the Wellcome Trust Sanger Institute for shotgun sequencing as follows: 226N16 (NCBI accession number CU695224), 243M20 (CU695226), 445J9 (CU695230), 332A4 (CU695228), 333C17 (CU695229), 240F21 (CU695225), 316B17 (CU695227) and 76B7 (CU695231). The sequences of IRG genes were extracted from these results and cross-checked with the data from the MGP. Clones containing IRG genes from the mouse strain MSM/Ms (*Abe et al., 2004*) were chosen based on the BAC end sequences from RIKEN BRC (http://www.brc.riken.jp/lab/dna/en/MSMBACen.html). Seven clones were sent to the Beijing Genomics Institute (BGI) for Illumina sequencing: 329H21, 362F04, 544P17, 494M12, 419B05, 148E20 and 355D01. The sequencing results were assembled and uploaded to the NCBI GenBank with accession number KF705682, KF705684, KF705686, KF705680, KF705685, KF705681, KF705683. IRG genes of the CIM mouse strain were sequenced via full transcriptome Illumina sequencing of IFNγ-induced diaphragm-derived cells (DDC, see below) in the Cologne Centre of Genomics (CCG). For wild mouse samples, genomic DNA was acquired from a variety of sources. From this material 5 key IRG members were amplified by PCR with appropriate primers (*Table 3*). The full ORFs of *Irgm1, Irgb2, Irgb6* and *Irgb10*, and a partial sequence of *Irga6* (964 bp in length) were amplified. PCR products were cloned into the pGEM-T vector and insertions in individual positive clones were sequenced. The sequences are attached as supplementary files in FASTA format.

## Bioinformatics and sequence analysis

Neighbor-joining trees of IRG genes and proteins were built with MEGA5 (*Tamura et al., 2011*). The MSM/Ms BAC Illumina data were assembled de novo with the assistance of Geneious Pro 5.5.6 (Biomatters Ltd.). Dot plots of genomic IRG gene clusters of BL/6 vs CIM and BL/6 vs MSM/Ms were calculated by LBDOT 1.0 (Lynnon Corporation) with a sliding window of 20 bp and a maximum mismatch of 2 bp. The raw Illumina reads from inbred mouse strains were accessed by LookSeq and all reads were grouped based on SNPs and manually aligned to their BL/6 homologues. Individual IRG gene sequences are available through Genbank. For analysis of the average diversity between house mouse strains, the sequences of 50 random functional genes were acquired from MGP and the National Institute of Genetics (NIG), Japan (http://molossinus.lab.nig.ac.jp/msmdb/). The analysis covered seven mouse strains: two laboratory inbred strains, BL/6 (MHC haplotype b) and AKR/J (MHC haplotype k); four wild-derived inbred house mouse strains, MSM/Ms, CAST/EiJ, PWK/PhJ and WSB/EiJ; the *M. spretus*

**Table 2.** IRG sequences from NCBI database

| Gene | Strain | Type | Access number |
|------|--------|------|---------------|
| *Irgm1* | Czech II | ESTs | BI150356, BF161711, BF168437, BF164781, BE367794 |
| *Irgb2-b1* | Czech II | mRNA | BC022776 |
| | JYG | mRNA | AK145236 |
| *Irgb6* | Czech II | mRNA | BC093522 |
| | Czech II | mRNA | BC034256 |
| | JYG | mRNA | AK166353 |
| | NMRI | mRNA | BC085259 |
| | C.D2 | mRNA | U15636 |
| *Irgd* | Czech II | mRNA | BC001986, BC009131 |
| *Irgm2* | Czech II | ESTs | BG518498, BF137080, BE284209, BF168033, BI149246, BI414397, BE283352, BE306442 |
| *Irgm3* | Czech II | ESTs | BI414397, BI149246, BF163420, BE283352, BF168033, BI153387, BE281683, BE306442, BI106672, BI150745, BF225799, BF168273 |
| *Irga6* | Czech II | ESTs | BF143764, BI150692, BF163277, BE369870, BI152144, BF168743, BF022265, BI105027, BE306549, BF140175 |
| | NMRI | ESTs | BG862486, BI654967, BI854263, BI654186, BG974278, BI662561, BI853679, BG864306, BI658908 |
| *Irga8* | Czech II | mRNA | BC023105 |
| *Irga9* | Czech II | mRNA | BC040796 |
| *Irga10* | Czech II | mRNA | BC020118 |

strain Spretus/EiJ. As shown in *Figure 3—source data 1*, genes were arbitrarily selected based on numerical position in the NCBI reference assembly build 37. Potentially functional genes closest to the designated genome position with EST evidence on the NCBI database were chosen. If the ORF of the gene was longer than 1500 bp, only 1500 bp were considered. Tajima's $\pi$ and $\pi a/\pi s$ values were calculated with DnaSP5 (*Librado and Rozas, 2009*). The secondary structure of Irgb2-b1 was predicted by PSIpred (UCL department of computer science, http://bioinf.cs.ucl.ac.uk/psipred/).

## Culture of *T. gondii* strains

*T. gondii* strains (*Table 4*) were maintained by serial passage in confluent monolayers of Hs27 cells. When Hs27 cells were lysed by *T. gondii* tachyzoites, parasites were harvested from the supernatant and purified from host cell debris by differential centrifugation (5 min at 100×*g*, 15 min at 500×*g*). The pelleted parasites were resuspended in IMDM, 5% FCS supplemented with 100 U/ml penicillin, 100 µg/ml streptomycin (PAA, Pasching, Austria), counted and immediately used for infection of mice, cells or lysed for subsequent immunoblot.

## Rationale for assay of specific IRG proteins in these experiments

The IRG proteins all have distinctive properties. Irga6 and Irgb6 are the most highly expressed (in lab mice) and there are excellent serological reagents available for them. We therefore used Irga6 and Irgb6 for experiments that monitor loading at the vacuole. In addition, we have antisera against the specific phosphoserines on Irga6 so we can measure phosphorylation directly. Irgb6 is however more drastically affected by the genetic difference between avirulent types II and III *T. gondii* strains and virulent type I strains, dropping from up to 90% loading of vacuoles to around 10%. Irga6 is also greatly affected but the effect is seen more conspicuously as a reduction in the intensity of loading rather than in a reduction in the percent of vacuoles detectably loaded. These and other distinctive properties, are described elsewhere (*Khaminets et al., 2010*; *Steinfeldt et al., 2010*; *Fleckenstein et al., 2012*). The tandem IRG protein, Irgb2-b1, became a focus of attention because of its high expression in resistant CIM mice, its large polymorphic variation, and evidence for being under recent divergent selection (see below).

**Table 3.** Primer list

| Name | Sequence 5′ to 3′ | Function |
|------|-------------------|----------|
| Irga6_56B_fw | CTACTATGAATGGTATATGTAGCATTGTG | *Irga6* amplification |
| Irga6_56B_bw | CAGGACTTCAGCTTAATTAGAAGGC | *Irga6* amplification |
| Irgb2_66F_f | CTGGACTCTGCGCTTTTATTGG | *Irgb2* amplification |
| Irgb2_66F_b | CTGGAAACACTTTGCCCACG | *Irgb2* amplification |
| Irgb6_67Y_fw | CCTCTCTTCTCCATTCAGCTTC | *Irgb6* amplification |
| Irgb6_67Y_bw | CCAAGGTGAAGCTAAGAGTGAAC | *Irgb6* amplification |
| Irgb10_682_fw | CTCCAGTGTCCTGTGTGCCC | *Irgb10* amplification |
| Irgb10_682_bw | CAGGAATGCCCTCAGTCGTC | *Irgb10* amplification |
| Irgm1_655_fw | CTGCCGATTCGATTCATAAAC | *Irgm1* amplification |
| Irgm1_655_bw | CCTCTCAGAGAATCTAAAACCC | *Irgm1* amplification |
| Irgm1_66F_bw | GAGACAGGGGAGATGAGTGAT | *Irgm1* amplification |
| Irga1_221_fw | ATCGATAGTTCCCTTGTCAATGTGG | backcross and F2 mice genotyping, Chr 18 |
| Irga1_221_bw | TTTGTAGAGTTTGGCTAGGGCCTG | backcross and F2 mice genotyping, Chr 18 |
| Irgb6_21D_fw | ATGGCTTGGGCCTCCAGCTT | backcross and F2 mice genotyping, Chr 11 |
| Irgb6_614_bw | CCACCATTCCACTTGGTGG | backcross and F2 mice genotyping, Chr 11 |
| Tox-9 | AGGAGAGATATCAGGACTGTAG | *T. gondii* qPCR primer |
| Tox-11 | GCGTCGTCTCGTCTAGATCG | *T. gondii* qPCR primer |
| Nlrp1_FW | AACTTATCTCAGGTCTCTGTGATT | Nlrp1b genotyping, forward |
| Nlrp1_BL6 | GATATAGGTCAGGACCAATGC | Nlrp1b backward, BL/6 specific |
| Nlrp1_CIM | GATATAGGTCAGGACCATCAA | Nlrp1b backward, CIM specific |

## Preparation of tissue culture lines from mouse diaphragm cells

The origins of mice used in this study are listed in *Table 5*. Cells were prepared from diaphragm tissue by a modification of the technique described by *Antony et al., (1989)*. Diaphragm-derived cells are easy to prepare and have the advantage over MEFs that they can be prepared from a single adult mouse, enabling individual genetically different animals to be studied genetically and functionally at the cellular level. One mouse from each of BL/6, CAST/EiJ, CTP and CIM strains was sacrificed and the diaphragm removed under sterile conditions. The diaphragm was washed with PBS, chopped up, incubated with collagenase/dispase (1 mg/ml, Roche, Mannheim, Germany) for 1 hr at 37°C and then

**Table 4.** *T. gondii* strains used in this study

| Type | Strain name | Reference | Note |
|------|-------------|-----------|------|
| I (virulent) | RH | (*Albert and Sabin, 1941*) | |
| | RH-YFP | (*Gubbels et al., 2003*) | transgenic RH strain expressing YFP |
| | RHΔ*rop5* | (*Behnke et al., 2011*) | transgenic RH strain, the *ROP5* locus has been deleted |
| | RHΔ*rop18* | (*Reese et al., 2011*) | transgenic RH strain, the *ROP18* locus has been deleted |
| | BK | (*Winsser et al., 1948*) | |
| | GT-1 | (*Dubey 1980*) | canonical type I strain, full sequence in ToxoDB Database |
| II (avirulent) | ME49 | (*Lunde and Jacobs, 1983*) | |
| III (avirulent) | NED | (*Darde et al., 1992*) | |

**Table 5.** Origin of mouse samples and mouse genomic DNA

| Sample name | Subspecies | Origin | Location of collection | Provided by |
|---|---|---|---|---|
| D9, D18, D12, D22, D31, D34 | *M. m. domesticus* | Germany | 50°50'N 6°45'E | Genomic DNA provided by B Harr Max Planck Institute for Evolutionary Biology, Germany |
| MC8, MC4, MC6, MC52, MC13, MC27, MC58 | *M. m. domesticus* | France | 44°20'N 3°0'E | |
| W1.1, W3.1, W3.2, W4.1, W7.1 | *M. m. musculus* | Austria | 48°12'N 16°22'E | |
| AL12, AL21, AL24, AL30, AL32, AL41 | *M. m. musculus* | Kazakhstan | 43°N 77°E | |
| MW2, MW4 | *M. m. musculus* | Inner Mongolia China | 41°5'N 108°9'E | Caught by J Lilue for this study. Institute for Genetics, University of Cologne, Germany |
| MT1, MT2 | | | 40°47'N 111°1'E | |
| JH4, JH6, JH11, JH12 | *M. m. musculus* or Hybrid zone | Hebei Province China | 37°37'N 115°19'E | |
| YX3, YX5, YX11 | *M. m. castaneus* | Henan Province China | 32°4'N 115°3'E | |
| MIB3, MIB4, MIB6, | *M. m. castaneus* | India | 13°3'N 77°34'E | Caught by UB Müller for this study. Institute for Genetics, University of Cologne, Germany |
| MIB23, MIB24, MIB25 | | | 13°6'N 77°34'E | |
| MIB35, MIB36 | | | 12°54'N 77°29'E | |
| CTP (living mice) | *M. m. castaneus* | Thailand | Mouse strain | F Bonhomme, Institut de Science de l'Evolution, Montpellier, France |
| CIM (living mice) | *M. m. castaneus* | India | | |
| CAST/Ei (living mice) | *M. m. castaneus* | Thailand | Inbred strain | The Jackson Laboratory, Bar Harbor, Maine, USA |
| C57BL/6 (living mice) | *M. m. domesticus* | Lab mouse | Inbred strain | Centre for Mouse Genetics, University of Cologne, Germany |
| NMRI (living mice) | *M. m. domesticus* | Lab mouse | Inbred strain | Charles River Laboratories, Sulzfeld, Germany |

centrifuged for 15 s at 100×*g*. The supernatant was collected, centrifuged for 5 min at 500×*g* and the pellet plated in DMEM, 10% FCS supplemented with 4 mM L-glutamine, 2 mM non-essential amino acids, 1 mM sodium pyruvate, 1× MEM non-essential amino acids, 100 U/ml penicillin, 100 μg/ml streptomycin (all PAA, Pasching, Austria). The remaining cell debris after collagenase/dispase-incubation was further incubated in 1× trypsin (Gibco, Grand Island, New York, USA) for 1 hr at 37°C, and then centrifuged for 15 s at 100×*g*. The supernatant was collected, centrifuged for 5 min at 500×*g* and the pellet plated (see above). Primary diaphragm-derived cells (DDC) were grown until they had reached ~50% confluence and then transfected with 2 μg of psv3-neo (*Southern and Berg, 1982*) using the FuGENE HD transfection reagent (Roche, Mannheim, Germany) according to the manufacturer's protocol. Cells were put under selection with G418 (Geneticin, PAA, Pasching, Austria) at a concentration of 150 μg/ml until immortalised clones had overgrown the culture. DDC isolated from mice and the mouse cell line L929 (from mouse strain C3H) were maintained in supplemented DMEM (see above) without G418.

## Cell induction, transfection and infection

Cells were induced for 24 hr with 200 U/ml of IFNγ (Peprotech, Rocky Hill, New York, USA) unless indicated otherwise. Irga6$_{BL/6}$ and Irga6$_{CIM}$ were cloned into the pGW1H vector with C-terminal ctag1 tags (*Martens et al., 2005*); full length Irgb2-b1$_{BL/6}$ and Irgb2-b1$_{CIM}$ were cloned into the pGW1H vector with C-terminal Flag tags. Constructs were transfected using FuGENE HD Transfection Reagent (Roche, Mannheim, Germany) according to the manufacturer's protocol. The multiplicities of infection (MOI) were 1 for the [3]H-uracil incorporation assay, 2–5 for immunofluorescence microscopy, 2.5, 5 and 10 for the cell viability assay, and ~10 for in-cell phosphorylation experiments. Cells were either fixed for immunofluorescence or lysed for western blot 2 hr after infection.

## Immunofluorescence microscopy and analysis

Cells were fixed with PBS/3% paraformaldehyde (PFA) for 20 min at room temperature (RT), washed three times with PBS and then permeabilized with 100% methanol on ice (for stainings including serum 87,558, below) or PBS/0.1% saponin at RT (all other antibodies) for 10 min followed by blocking with PBS/3% bovine serum albumin (BSA) for 1 hr. Cells were incubated with primary antibodies diluted in PBS/3% BSA for 1 hr and subsequently incubated with secondary antibodies for 30 min at RT. Antibodies against Irgb6 (141/1) and against the conserved Irgb-tandem C-terminal peptide CLSDLPEYWETGMEL (954/1-C15A) shared by Irgb-tandem proteins of both BL/6 and CIM mice raised at Innovagen AB (Lund, Sweden). Rabbit polyclonal anti-Irga6 phosphorylated at T108 has been described (serum 87,558) (*Steinfeldt et al., 2010*). Other primary immunoreagents were anti-recombinant Irga6 antiserum 165/3 (*Martens et al., 2004*), mouse anti-FLAG (M2, Sigma-Aldrich, St. Louis, Missouri, USA) and rabbit anti-calnexin (Calbiochem, Darmstadt, Germany). Second-stage antibodies were: Alexa 488 and Alexa 555 labelled donkey anti-mouse and anti-rabbit sera (Molecular Probes, Eugene, Oregon, USA). Images were taken with a Zeiss Axioplan II fluorescence microscope equipped with an AxioCam MRm camera (Zeiss, Jena, Germany). Images were processed with Axiovision 4.7 (Zeiss, Jena, Germany). Quantification of IRG protein signal intensity at the *T. gondii* PVM was performed as described before (*Khaminets et al., 2010*). All quantification of microscopical images was performed double blind. Error bars in *Figure 6A,G* represent standard deviations of repeated measurements.

## Western blot analysis

$4 \times 10^5$ cells were seeded to individual wells of a six-well plate and induced with IFNγ for 24 hr. Cell lysis and western blot analysis was performed essentially as described elsewhere (*Steinfeldt et al., 2010*).

## Immunoprecipitation

$6 \times 10^5$ BL/6 and CIM DDC were seeded in 6-cm dishes and induced with IFNγ for 24 hr. Metabolic labelling with $^{33}$P-phosphoric acid (Hartmann Analytic, Braunschweig, Germany), cell lysis and immunoprecipitation was performed essentially as described elsewhere (*Steinfeldt et al., 2010*).

## In vitro $^3$H-uracil incorporation assay for measuring *T. gondii* proliferation

*T. gondii* proliferation was measured using the $^3$H-uracil incorporation assay (*Pfefferkorn and Guyre, 1984*). DDC were seeded on 96-well plates (6500 cells/well) and induced with IFNγ (100 U/ml) or left untreated. 24 hr after induction cells were infected for a further 24 hr with specified *T. gondii* strains at different multiplicities of infection, or left uninfected. The cultures were labeled with 0.3 µCi/well of $^3$H-uracil ($^3$HU, Hartmann Analytic, Braunschweig, Germany) for 24 hr and then frozen at −20°C. The amount of radioactivity incorporated into proliferating parasites was determined by a MatrixTM 9600 β-counter (Packard, Meriden, Connecticut, USA). Data are shown for MOI = 1 and presented as the percentage of residual parasite proliferation under IFNγ treatment (*Figure 4B*). Residual parasite proliferation was defined as follows: 100—([$^3$HU counts—background in infected, IFNγ-treated culture/mean $^3$HU counts—background in infected, non-treated cultures] ×100) where background is $^3$HU (mean) counts of uninfected, non-induced cultures.

## Cell viability assay

DDC were seeded and induced as described for the $^3$H-uracil incorporation assay (see above). Cells were infected with type I RH-YFP or type II ME49 strains of *T. gondii* with indicated MOIs for 8 hr. Viable cells were quantified by the CellTiter 96 AQ$_{ueous}$ non-radioactive cell proliferation assay (Promega, Madison, Wisconsin, USA) according to the manufacturer's protocol. The absorption of a bioreduced formazan of the tetrazolium compound MTS, which is generated by metabolically active cells during incubation at 37°C for 2–4 hr, was measured in an ELISA reader (Molecular Devices, Menlo Park, California, USA) at 490 nm. The quantity of formazan product is proportional to the number of living cells in the culture.

## In vivo survival assay and genotyping

Mice were infected i.p. with 500 RH-YFP tachyzoites in 200 µl of PBS and tail samples were taken when animals succumbed during the acute phase of infection. Survivors were sacrificed 60 days post infection, tested for sero-conversion using the Toxocell Latex Kit (biokit, Barcelona, Spain) and tail biopsies taken. Biopsies were digested in 500 µl of buffer (100 mM Tris-HCl [pH 8.5], 5 mM EDTA, 200 mM NaCl, 0.2% SDS, 150 µg/ml proteinase K) and genomic DNA precipitated with isopropanol. An 804 bp fragment

of *Irga1* and an 857 bp fragment of *Irgb6* were amplified from genomic DNA using the primers listed in *Table 3*. PCR products were digested with restriction enzymes AccI (*Irga1*) or FokI (*Irga6*, both New England BioLabs, Ipswich, Massachusetts, USA) for 45 min at 37°C, followed by 20 min at 60°C. DNA fragments were separated on a 2% agarose gel. *Nlrp1b* fragments were amplified with a universal forward primer and strain-specific backward primers (BL/6 or CIM, see *Table 3*).

## Flow cytometry to assay infected peritoneal cells

Mice were infected i.p. with 500 RH-YFP tachyzoites, sacrificed on day 5 post infection and subsequently subjected to peritoneal lavage with 6 ml of PBS. Lavage suspension (1 ml) was centrifuged in microtubes for 5 min at 500×$g$, the supernatant was discarded, the cell pellet resuspended with 30 µl of PBS/0.5% BSA containing PE-conjugated rat anti-mouse CD45 antibody (BD Biosciences, San Jose, California, USA) and subsequently incubated for 20 min on 4°C in the dark. The microtubes were then filled with PBS/0.5% BSA, centrifuged for 5 min at 500×$g$, the supernatant discarded and stained cells resuspended in 300 µl of PBS/0.5% BSA. Cells were analysed using a BD FACS Calibur flow cytometer (BD Biosciences, San Jose, California, USA) and the percentage of infected CD45-positive cells was calculated as percentage of events positive for YFP over 5×$10^4$ CD45-positive cells using WinMDI 2.9.

## Quantitation of *T. gondii* in brains of infected mice

Mice were infected with *T. gondii* tachyzoites in 200 µl of PBS as indicated in *Table 1*. 6–8 weeks (GT-1 and BK) or 5 weeks (NED) post infection mice were sacrificed, the brains removed and triturated in 1 ml of PBS. Cysts were counted in 15–20 drops of 10 µl per brain homogenate to estimate the total number of cysts per brain. Additionally, homogenised mouse brains were digested with proteinase K (final concentration 100 µg/ml) overnight, and total genomic DNA was isolated with DNeasy Blood & Tissue Kit (Qiagen, Hilden, Germany) according to the manufacturer's protocol. The presence of *T. gondii* DNA was detected with a Taqman qPCR method (7900fast; Applied Biosystems, Foster City, California, USA) by primer Tox-9 and Tox-11 as described before (*Reischl et al., 2003*).

## Statistical analysis

Differences were tested for statistical significance using the unpaired two-tailed Student's *t* test.

## Acknowledgements

We thank Claudia Poschner for outstanding technical support. Stephanie Könen-Waisman helped with double-blind evaluation of IRG protein loading onto *T. gondii* vacuoles. We thank Miriam Linnenbrink and Christine Pfeifle (MPI Evolutionary Biology, Plön) for expertise in handling wild mice, and Christine Pfeifle for maintaining the CIM strain in Plön. Annie Orth and François Bonhomme (U Montpellier) provided the CIM breeding colony. Uma Ramakrishnan and colleagues at the NCBI, Bangalore provided the facilities to prepare cells from locally caught wild mice. Olaf Utermoehlen (U Cologne) provided facilities for the infection of wild mice with *T. gondii*. Carsten Lüder (U Göttingen) and Ildiko Dunay (U Magdeburg) advised on identification and analysis of *T. gondii* cysts in vivo. Aurélien Tellier (LMU, Munich) and Stephan Schiffels (U Cologne) contributed to analysis of the sequence alignments. We are grateful to Paul Schulze-Lefert (Cologne), Paul Schmid-Hempel (Zurich) and Isabel Gordo (Oeiras) for comments on the manuscript.

## Additional information

### Funding

| Funder | Grant reference number | Author |
| --- | --- | --- |
| Deutsche Forschungsgemeinschaft | SFB 680, SFB 635, SFB 670 and SPP 1399 | Jingtao Lilue, Urs Benedikt Müller, Tobias Steinfeldt, Jonathan C Howard |
| International Graduate School in Development Health and Disease | | Urs Benedikt Müller |

The funders had no role in study design, data collection and interpretation, or the decision to submit the work for publication.

## Author contributions

JL, UBM, TS, Conception and design, Acquisition of data, Analysis and interpretation of data, Drafting or revising the article; JCH, Conception and design, Analysis and interpretation of data, Drafting or revising the article

## Ethics

Animal experimentation: All animal experiments were conducted under the regulations and protocols for animal experimentation by the local government authorities (Bezirksregierung Köln, Germany), LANOV Nordrhein-Westfalen Permit No. 44.07.189.

# Additional files

## Major dataset

The following datasets were generated:

| Author(s) | Year | Dataset title | Dataset ID and/or URL | Database, license, and accessibility information |
| --- | --- | --- | --- | --- |
| Lilue J, Müller UB, Steinfeldt T, Howard JC | 2013 | Mus musculus molossinus clone MSMg01-329H21 | KF705682; http://www.ncbi.nlm.nih.gov/nuccore/?term=KF705682 | Publicly available at GenBank (http://www.ncbi.nlm.nih.gov/genbank/). |
| Lilue J, Müller UB, Steinfeldt T, Howard JC | 2013 | Mus musculus molossinus clone MSMg01-362F04 | KF705684; http://www.ncbi.nlm.nih.gov/nuccore/?term=KF705684 | Publicly available at GenBank (http://www.ncbi.nlm.nih.gov/genbank/). |
| Lilue J, Müller UB, Steinfeldt T, Howard JC | 2013 | Mus musculus molossinus clone MSMg01-544P17 | KF705686; http://www.ncbi.nlm.nih.gov/nuccore/?term=KF705686 | Publicly available at GenBank (http://www.ncbi.nlm.nih.gov/genbank/). |
| Lilue J, Müller UB, Steinfeldt T, Howard JC | 2013 | Mus musculus molossinus clone MSMg01-494M12 | KF705680; http://www.ncbi.nlm.nih.gov/nuccore/?term=KF705680 | Publicly available at GenBank (http://www.ncbi.nlm.nih.gov/genbank/). |
| Lilue J, Müller UB, Steinfeldt T, Howard JC | 2013 | Mus musculus molossinus clone MSMg01-419B05 | KF705685; http://www.ncbi.nlm.nih.gov/nuccore/?term=KF705685 | Publicly available at GenBank (http://www.ncbi.nlm.nih.gov/genbank/). |
| Lilue J, Müller UB, Steinfeldt T, Howard JC | 2013 | Mus musculus molossinus strain MSM/Ms clone MSMg01-148E20 | KF705681; http://www.ncbi.nlm.nih.gov/nuccore/?term=KF705681 | Publicly available at GenBank (http://www.ncbi.nlm.nih.gov/genbank/). |
| Lilue J, Müller UB, Steinfeldt T, Howard JC | 2013 | Mus musculus molossinus clone MSMg01-355D01 | KF705683; http://www.ncbi.nlm.nih.gov/nuccore/?term=KF705683 | Publicly available at GenBank (http://www.ncbi.nlm.nih.gov/genbank/). |

The following previously published datasets were used:

| Author(s) | Year | Dataset title | Dataset ID and/or URL | Database, license, and accessibility information |
| --- | --- | --- | --- | --- |
| Yalcin B, Adams DJ, Flint J, Keane TM | 2012 | Mouse Genomes Project | http://www.sanger.ac.uk/resources/mouse/genomes/ | These data are released in accordance with the Fort Lauderdale agreement and Toronto agreements. |
| Abe K, Noguchi H, Tagawa K, Yuzuriha M, Toyoda A, Kojima T, Ezawa K, Saitou N, Hattori M, Sakaki Y, Moriwaki K, Shiroishi T | 2004 | NIG Mouse Genome Database | http://molossinus.lab.nig.ac.jp/msmdb/ | Available at NIG Mammalian Genetics Laboratory, Japan. |
| Holt K | 2008 | Mus musculus castaneus strain CAST/Ei clone CH26-332A4 | CU695228; http://www.ncbi.nlm.nih.gov/nuccore/CU695228 | Publicly available at the NCBI Nucleotide database (http://www.ncbi.nlm.nih.gov/nuccore). |
| Holt K | 2008 | Mus musculus castaneus strain CAST/Ei clone CH26-226N16 | CU695224; http://www.ncbi.nlm.nih.gov/nuccore/CU695224 | Publicly available at the NCBI Nucleotide database (http://www.ncbi.nlm.nih.gov/nuccore). |

| Holt K | 2008 | Mouse DNA sequence from clone CH26-243M20 on chromosome 11, complete sequence | CU695226; www.ncbi. nlm.nih.gov/nuccore/ CU695226 | Publicly available at the NCBI Nucleotide database (http:// www.ncbi.nlm.nih.gov/ nuccore). |
|--------|------|---------------------------------------------------------------------------------|-----------------------------------------------------------|-----------------------------------------------------------------------------------------|
| Holt K | 2008 | Mouse DNA sequence from clone CH26-455J9 on chromosome 11, complete sequence | CU695230; http://www.ncbi. nlm.nih.gov/nuccore/ CU695230 | Publicly available at the NCBI Nucleotide database (http:// www.ncbi.nlm.nih.gov/ nuccore). |
| Holt K | 2008 | Mouse DNA sequence from clone CH26-333C17 on chromosome 11, complete sequence | CU695229; http://www.ncbi. nlm.nih.gov/nuccore/ CU695229 | Publicly available at the NCBI Nucleotide database (http:// www.ncbi.nlm.nih.gov/ nuccore). |
| Holt K | 2008 | Mus musculus castaneus strain CAST/Ei clone CH26-240F21 | CU695225; http://www.ncbi. nlm.nih.gov/nuccore/ CU695225 | Publicly available at the NCBI Nucleotide database (http:// www.ncbi.nlm.nih.gov/ nuccore). |
| Holt K | 2008 | Mouse DNA sequence from clone CH26-316B17 on chromosome 18, complete sequence | CU695227; http://www.ncbi. nlm.nih.gov/nuccore/ CU695227 | Publicly available at the NCBI Nucleotide database (http:// www.ncbi.nlm.nih.gov/ nuccore). |
| Holt K | 2008 | Mouse DNA sequence from clone CH26-76B7 on chromosome 18, complete sequence | CU695231; www.ncbi.nlm.nih. gov/nuccore/CU695231 | Publicly available at the NCBI Nucleotide database (http:// www.ncbi.nlm.nih.gov/ nuccore). |

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
