## [Decision Letter]

Thank you for sending your work entitled “Toxoplasma and the mouse: reciprocal virulence and resistance polymorphism” for consideration at *eLife*. Your article has been favorably peer reviewed by Detlef Weigel, *eLife* Deputy editor, and two other reviewers.

The Deputy editor and the other reviewers discussed their comments before we reached this decision, and the Deputy editor has assembled the following comments to help you prepare a revised submission.

This manuscript reports on the role of different mouse *IRG* genotypes in host defense against *Toxoplasma gondii*. The complexity and level of polymorphism in the IRG system are comparable to that at the MHC, and in a wider collection of mouse strains *IRG* genotypes can be found that confer resistance to strains of *T. gondii* that are highly virulent in laboratory mouse strains. In addition, the authors provide evidence that specific *IRGs* from resistant mouse strains function by dysregulating the activity of parasite virulence factors. The experiments are well designed and conducted, and the results are properly interpreted. The authors go on to speculate that the *IRG* polymorphism in mice (and loss of the entire IRG system in several vertebrate groups) reflects a fitness trade-off. In addition, the authors acknowledge that the mouse–*T. gondii* interaction is probably only a part of the evolutionary picture, and that *IRG* evolution is likely to be driven by other pathogens as well, such as Chlamydia.

The reviewers, however, felt that some of the evolutionary inferences are overstated. For example, the abstract begins with “Virulence in *T. gondii* for its natural intermediate hosts, the mouse, is an evolutionary paradox…” This would only be the case if one takes laboratory mice, known to be highly derived, as representative of wild mice. In fact, prior studies have shown that deer mice (*Peromyscus*) are resistant to virulent strains of *T. gondii* (Am. J. Trop. Med. Hyg. 2, 390-415 (1953), as is the case for other common hosts such as chickens and rats. It is therefore reasonable to assume that wild house mice might express resistance mechanisms not found in laboratory mice, given the extremely narrow genetic diversity of the latter. The authors find such a mechanism and this is certainly interesting, but perhaps not as surprising as claimed. Hence, one could conclude that the virulence mechanisms of the parasite have evolved for “resistant hosts” in general and not wild mice per se. Although the system seems to resemble the R gene-avirulence gene systems for plants and their pathogens, the co-evolution arguments should be toned down considerably, as the work stands on its own.

Major comments:

1) There is a concern about the mechanism proposed for Irgb2-b1 function. In cells from CIM mice infected with RH parasites, the authors demonstrate a substantial increase in Irgb6 and Irgb2-b1 loading onto the parasite vacuole, compared to cells from B6 mice. In addition, although there is undetectable p-Irga6_CIM_ in CIM cells infected with RH parasites, Irga6_CIM_ can be phosphorylated if transfected into other cells (L929 cells), suggesting that Irga6 phosphorylation is actively inhibited in CIM cells. The authors state that Irgb2-b1_CIM_ inhibits the phosphorylation of endogenous Irga6 in CIM mice. This statement is based on experiments in which they transfect Irgb2-b1_CIM_ into B6 cells stimulated with IFN-γ and observe a reduction in p-Irga6 localization to the parasite vacuole membrane, which they quantify. However, they do not directly show that Irgb2-b1_CIM_ inhibits Irga6 phosphorylation. It seems that an alternate interpretation of these data (from Figure 6) is that Irgb2-b1_CIM_ may inhibit the localization of Irga6 to the vacuole, where it is then phosphorylated. This point should be addressed/clarified.

Overall, the data implicating ROP5 in the loading of Irgb2-b1_CIM_ onto the vacuole surrounding virulent parasites is very indirect. Granted it is decreased in Δrop5 parasites, but this does not equate to a model where Irgb2-b1 binds to ROP5 or inhibits its activity. Other indirect mechanisms are possible. This portion of the manuscript is too speculative. Either concrete data need to be provided or this should be extensively modified to indicate it might result from direct or indirect binding. Does Irgb2-b1 have a direct effect on ROP18? It would look the same phenotypically (*Δrop5* parasites would phenocopy this). The model promoted here suggests that ROP5 binds to IRGs. If this interaction is robust, it should be straightforward to demonstrate this for Irgb2-b1. If such data are elusive, the authors should probably rethink their model. Figure 7 is somewhat speculative, and is also not well explained. For example, what is the significance of the IIa/IIs ratios in the graphic?

In the last paragraph of the Results when the authors describe Figure 7, they state, “Preliminary results suggest that Irgb2-b1_CIM_ may bind directly to the protein product of the virulent allele of ROP5.” What are these preliminary results? Including this interaction data would significantly strengthen the authors’ model that Irgb2-b1_CIM_ binding to ROP5 may contribute to blocking Irga6 phosphorylation in CIM mice.

2) The authors rely at least in part on Illumina resequencing data for their phylogenetic studies; such data are notoriously problematic for highly polymorphic and complex, duplicated regions of the genome. It would be appropriate for the authors to discuss how the BAC sequences they generated (which are only mentioned in the Materials and methods) as well as their extensive PCR data compared with the Illumina resequencing data.

3) It might be worth pointing out that type I strains are relatively rare, yet they share the “acute lab mouse” phenotype with many South American strains that not coincidentally share a ROP18 allele (and also ROP5 although this is less well characterized). If it were only clonal type I strains at play, the evolutionary argument that natural hosts must somehow be resistant to such strains would be much less meaningful (type 1 strains are less than 5% of all strains encountered in the wild).

4) The extreme polymorphism of the IRG family of proteins suggests that other pathogens might also impinge on this pathway. This is only briefly alluded to in the Discussion, but in fact it seems relevant from the opening and should be mentioned in the Introduction already. The present work reveals the role of one component of the IRG family in combating Tg. And yet, the role of many other components remains undefined. Again, how important co-evolution with *T. gondii* is for *IRG* diversification is still unknown. Along these lines, the observation that the virulent strain GT1 can form cysts in resistant CIM mice is interesting, but not a very strong argument for co-evolution. Virulent strains of *T. gondii* can form cysts in any resistant species. In this regard, there does not seem anything particularly different about the wild mouse. It is the lab mouse that is the exception. It remains an interesting historical artefact that the extreme susceptibility of the lab mouse led to ROP kinase discovery. Along these lines, little is said about the coincidence between different *IRG* genotypes and *T. gondii* strains in the wild, which is obviously important to determine the significance of the laboratory observations.

5) The life cycle cartoon contains several errors. Oocysts contain two sporozoites each with four sporozoites, not two as shown. Tachyzoites are shown outside cells but should be replicating within them. Asexual stages also occur in the cat.

---

## [Author Response]

*The reviewers […] felt that some of the evolutionary inferences are overstated. For example, the abstract begins with “Virulence in* T. gondii *for its natural intermediate hosts, the mouse, is an evolutionary paradox…*”

Well, paradoxes are there to be resolved. While there is a literature with which we are familiar, and now cited, showing that individuals of other species (including man, of course) are resistant to many virulent *T. gondii* strains, the belief is general and has been reiterated for years, that mice are peculiarly vulnerable to the parasite. This generalisation has however been based on a limited database: until our own study, which is now pretty well known in the community, everybody has used classical laboratory mouse strains to examine this question. Yes, it's pretty obvious, but as said, (**A**) it can't be found in the published literature and (**B**) paradoxes are there to be resolved: this paper presents experimental data that resolve the paradox. Incidentally it is far from obvious that the laboratory strains of mouse should be so homogeneous in the *IRG* gene clusters. The MHCs of the same strains are strikingly polymorphic. Their shared vulnerability to type I Toxoplasma strains is the case simply because they have homogenised specific haplotypes of the IRG system. One may wonder whether this is due to an undiscovered fitness cost of certain *IRG* haplotypes rather than to a limited genetic basis among the lab strains.

*This would only be the case if one takes laboratory mice, known to be highly derived, as representative of wild mice. In fact, prior studies have shown that deer mice (Peromyscus) are resistant to virulent strains of* T. gondii *(Am. J. Trop. Med. Hyg. 2, 390-415 (1953), as is the case for other common hosts such as chickens and rats. It is therefore reasonable to assume that wild house mice might express resistance mechanisms not found in laboratory mice, given the extremely narrow genetic diversity of the latter. The authors find such a mechanism and this is certainly interesting, but perhaps not as surprising as claimed*.

We do not say anywhere that it is surprising: indeed it was the original apparently general “susceptibility” that was surprising, that was the “paradox”.

*Hence, one could conclude that the virulence mechanisms of the parasite have evolved for “resistant hosts” in general and not wild mice per se*.

Not sure we understand this statement. Considering what virulence and resistance consist of, i.e. specific interactions between polymorphic molecules of host and pathogen, we do not really grasp what is meant by “‘resistant hosts’ in general”. As we show in Figure 2, there are multiple alleles for relevant resistance molecules: we have explored the resistance properties of a minute sample of the available allelic variation against a minute sample of virulence alleles. It is plausible that further resistance variants may confer protection against further virulence variants.

*Although the system seems to resemble the R gene-avirulence gene systems for plants and their pathogens, the co-evolution arguments should be toned down considerably, as the work stands on its own*.

We have lowered the prominence of our remarks on plant R-genes. Still, it is a conjecture based on properties the two systems have in common, but not implausible and good to think about, we find. To instance some of the similarities: both are encoded in short, uninterrupted clusters; both are deeply polymorphic; both show evidence of dynamic genomic behaviour within clusters; both show simple resistance/susceptibility behaviour confronted with strain-specific pathogen variants. The MHC shows these properties too, but MHC-related associations with infectious disease resistance and susceptibility are usually much less clear-cut, for a number of reasons.

In summary, we appreciate the various reservations raised above and have made some modifications to the text. Nevertheless the views are ours and, unlike the experimental results themselves (hopefully), can be strengthened or weakened by future experiments. We feel that we should be entitled to express our views clearly, so long as they are not evidently wrong, even if a referee is not of the same opinion.

*1) There is a concern about the mechanism proposed for Irgb2-b1 function. In cells from CIM mice infected with RH parasites, the authors demonstrate a substantial increase in Irgb6 and Irgb2-b1 loading onto the parasite vacuole, compared to cells from B6 mice. In addition, although there is undetectable p-Irga6*_*CIM*_
*in CIM cells infected with RH parasites, Irga6*_*CIM*_
*can be phosphorylated if transfected into other cells (L929 cells), suggesting that Irga6 phosphorylation is actively inhibited in CIM cells. The authors state that Irgb2-b1*_*CIM*_
*inhibits the phosphorylation of endogenous Irga6 in CIM mice. This statement is based on experiments in which they transfect Irgb2-b1*_*CIM*_
*into B6 cells stimulated with IFN-γ and observe a reduction in p-Irga6 localization to the parasite vacuole membrane, which they quantify. However, they do not directly show that Irgb2-b1*_*CIM*_
*inhibits Irga6 phosphorylation. It seems that an alternate interpretation of these data (from Figure 6) is that Irgb2-b1*_*CIM*_
*may inhibit the localization of Irga6 to the vacuole, where it is then phosphorylated. This point should be addressed/clarified*.

The referee is correct, the data presented is open to another interpretation; thanks for pointing it out. Indeed we already have the data, which we now present as an additional image in Figure 6, that tests the alternative hypothesis suggested by the referee. In the images originally presented we showed only the deficit of p-Irga6 at vacuoles loaded with Irgb2-b1_CIM_. However using an antibody against total Irga6 we also measured the loading of total Irga6 at vacuoles loaded with Irgb2-b1_CIM_. New Figure 6 shows that, so far from being reduced as predicted by the referee's hypothesis, the loading of un-phosphorylated, and thus active, Irga6 is significantly increased by the presence of Irgb2-b1_CIM_. This result is expected if the absence of p-Irga6 shown in Figure 6 is because the Irgb2-b1_CIM_ acts, as proposed, as a decoy for the parasite kinase complex, drawing it away from its intended target, rather than as an inhibitor of Irga6 binding. Note that binding of Irgb6 at the vacuole is also increased by the presence of Irgb2-b1_CIM_ (Figure 6).

*Overall, the data implicating ROP5 in the loading of Irgb2-b1*_*CIM*_
*onto the vacuole surrounding virulent parasites is very indirect. Granted it is decreased in Δrop5 parasites, but this does not equate to a model where Irgb2-b1 binds to ROP5 or inhibits its activity. Other indirect mechanisms are possible. This portion of the manuscript is too speculative. Either concrete data need to be provided or this should be extensively modified to indicate it might result from direct or indirect binding. Does Irgb2-b1 have a direct effect on ROP18? It would look the same phenotypically (Δrop5 parasites would phenocopy this). The model promoted here suggests that ROP5 binds to IRGs. If this interaction is robust, it should be straightforward to demonstrate this for Irgb2-b1*.

The referee is correct that the evidence we present is indirect, though it is internally consistent, and externally consistent with the already published evidence for the direct interaction between virulent ROP5 and Irga6, with a precise structural basis, and shortly to be published as a crystal structure of the binary complex (from the Boothroyd lab and shortly to be published as a crystal structure of the binary complex [from Michael Reese and John Boothroyd's lab, PDB codes 4LV5 (ROP5Bi:Irga6) and 4LV8 (ROP5Ci:Irga6) to be released 25/09/2013]). It is important to stress that the IRG proteins are a homologous family and there is every reason to expect that they have homologous behaviour at a structural level. The participation of helix 4 of Irga6 in this interaction is thus also consistent with the evidence we present that helix 4 of Irgb2-b1 is under divergent selection (Figure 7), if Irgb2-b1 acts as a competitor of Irga6 for ROP5 and interacts directly with the virulence factor. The analysis of the mechanism by which Irgb-2b1 inhibits phosphorylation of Irga6 is not the main theme of this paper. On the basis of our results we consider that a direct interaction is the likeliest hypothesis, but a fuller analysis is beyond the scope of this paper. While these results are suggestive, further experimental analysis is needed to establish whether Irgb2-b1 indeed interacts directly with ROP5.

*If such data are elusive, the authors should probably rethink their model*.

To be clear, this is not yet a model; it is our present interpretation of the data and we consider it to be the most parsimonious. Certainly other interpretations are possible. The detail of the inhibitory mechanism is not central to the main argument of our paper, though of course is of great interest in itself.

*Figure 7 is somewhat speculative, and is also not well explained. For example, what is the significance of the IIa/IIs ratios in the graphic*?

The legend of Figure 7 has been modified to explain this data more clearly. Here is a more detailed account:

The πa/πs ratio gives an indication of the frequency with which coding variants occur in a sequence compared with the frequency of non-coding variants. It is related to the dN/dS parameter often used. The analysis is based on a moving window along the sequence and the colour coding records the regions with different frequencies of coding substitutions. The extended “red” domain on and adjacent to helix 4 identifies a region that appears to be under recent positive selection for a divergent sequence. This would occur, e.g., if this region of the molecule was under pressure to interact with new variants of a pathogen protein. Normally one would not expect to see this effect if Irgb2-b1 interacts with a host protein to “bridge” to ROP5 because the interaction surface between two host proteins would not be expected to evolve rapidly under divergent selection. We have added an additional comment on this to the text.

*In the last paragraph of the Results when the authors describe Figure 7, they state, “Preliminary results suggest that Irgb2-b1*_*CIM*_
*may bind directly to the protein product of the virulent allele of ROP5.” What are these preliminary results? Including this interaction data would significantly strengthen the authors’ model that Irgb2-b1*_*CIM*_
*binding to ROP5 may contribute to blocking Irga6 phosphorylation in CIM mice*.

The “preliminary results” are those specified in the paper and shown in Figure 7. We are working on this issue at the biochemical level as part of a further study on the details of the mechanism by which Irgb2-b1 interferes with Irga6 phosphorylation. As stated above, we feel we have made a good indirect case for a direct interaction with the ROP5 & ROP18 kinase complex, but a conclusive test is not straightforward. We consider it outside the scope of our paper and hope that the editors are prepared to leave this as it is.

*2) The authors rely at least in part on Illumina resequencing data for their phylogenetic studies; such data are notoriously problematic for highly polymorphic and complex, duplicated regions of the genome. It would be appropriate for the authors to discuss how the BAC sequences they generated (which are only mentioned in the Materials and methods) as well as their extensive PCR data compared with the Illumina resequencing data*.

To answer the referees in short, the Illumina resequencing is exceptionally reliable. To be clear, however, the only Illumina-derived transcriptome data of our own (at CCG Cologne) that we present here is the CIM transcriptome from IFNγ-induced DDC. The MSM BAC was sequenced by Illumina at Beijing Genomics Institute, de novo assembled then aligned to the C57BL/6 canonical genome. To test the validity of using Illumina resequencing of complete transcriptomes as a general approach for the rapid determination of unknown *IRG* haplotypes, we made an Illumina run on RNA of IFNγ-induced C57BL/6 fibroblasts to compare with the canonical genomic sequence of this strain that was established by extended Sanger sequencing on shotgun genomic fragments and from BACs. The C57BL/6 haplotype on Chr 11 is peculiarly difficult because it contains 4 very recently divergent tandem Irgb genes and additionally a duplicated Irgb6 gene, the two isoforms differing by only 3 non-coding nucleotides in the ORF. The resequencing results were not only easy to align against the canonical sequence, but also acted as a welcome confirmation of the sequences of the C57BL/6 genome: it was unclear for earlier releases of the C57BL/6 genome whether indeed there was an additional duplication in the IRGB region of Chr 11. The Illumina results show beyond doubt that both the extra copy of Irgb6 and the two extra tandem Irgb genes are real. All the intact *IRG* genes were recovered in up to 12,000 fold coverage; only degraded pseudogenes were not found. A second comparison was made with the CAST/Ei sequence. In that case, the data from the Sanger mouse genome project was based on Illumina resequencing while our data was based on Sanger sequencing of shotgun genomic fragments from BACs. Again there is perfect conformity between the two sequence sets.

Inevitably, Illumina resequencing of a transcriptome is unable to establish haplotypic linkage structure in heterozygotes carrying two *IRG* haplotypes, as is common in wild mice. The Sanger genomic Illumina data presented in this paper, however, is all obtained from inbred mice, whether of wild origin or laboratory strains.

Our resequencing reads and alignments are available if the referees would like to confirm their quality for themselves. The strength of the data is not only its consistency, but also the extreme depth of coverage, enabling us to identify and fully sequence *IRG* genes with expression levels over a 50x range or more. This favourable sensitivity is presumably achieved by the massive induction of *IRG* genes by IFNγ: we collect RNA from cells stimulated for 24 hr with high concentrations of IFNγ. At this time, *IRG* gene sequences may reach 1.0% of the total transcriptome.

*3) It might be worth pointing out that type I strains are relatively rare, yet they share the “acute lab mouse” phenotype with many South American strains that not coincidentally share a ROP18 allele (and also ROP5 although this is less well characterized). If it were only clonal type I strains at play, the evolutionary argument that natural hosts must somehow be resistant to such strains would be much less meaningful (type 1 strains are less than 5% of all strains encountered in the wild)*.

The referees are correct that Type I strains are rather uncommon in much of the Eurasian domain; however, they appear to be relatively abundant in some parts of the Far East, e.g., (Puvanesuaran, et al. 2013) Malaysia, 100% type I; (Kyan, et al. 2012) Japan, 6/14 type I; (Zakimi, et al. 2006) Japan 22/49 type I. It is perhaps relevant that the highly resistant and very similar *IRG* haplotypes of the CIM and CAST/Ei strains are both from the *Mm castaneus* subspecies local to Southeast Asia. We feel, however, that to stress such correlations on such limited data may encourage various speculations when what is needed is a detailed ecological analysis from the wild environment that we are presently undertaking, coupled with direct experimental support.

*4) The extreme polymorphism of the* IRG *family of proteins suggests that other pathogens might also impinge on this pathway. This is only briefly alluded to in the Discussion, but in fact it seems relevant from the opening and should be mentioned in the Introduction already*.

With all respect for the referees’ view, we feel we deal with this important point quite comprehensively, devoting more than a third of the entire Discussion to it. It is a very interesting issue, although it does not bear directly on the results that we present in this paper. It is clear from the data that polymorphism in at least one of the IRG proteins on chromosome 11 is key to control of *Toxoplasma* virulence. In our view it is premature to speculate intensively on what might be the adaptive meaning of the rest of the polymorphism. Maybe it has to do with other *Toxoplasma* virulence genotypes; maybe with resistance properties of other pathogen species. Certainly the complete set of mouse pathogens under IRG control is not yet known, but equally certainly it is a surprisingly small subset of the entire range.

Unless the editors insist, we would rather not raise this very interesting but in our view secondary point in the Introduction when we devote a considerable amount of space to it in the Discussion.

*The present work reveals the role of one component of the* IRG *family in combating Tg. And yet, the role of many other components remains undefined. Again, how important co-evolution with T. gondii is for IRG diversification is still unknown. Along these lines, the observation that the virulent strain GT1* can *form cysts in resistant CIM mice is interesting, but not a very strong argument for co-evolution*.

Agreed. Still, if only certain wild haplotypes are permissive for encystment by “virulent” strains, this does have, let us say, co-evolutionary potential.

*Virulent strains of* T. gondii *can form cysts in any resistant species. In this regard, there does not seem anything particularly different about the wild mouse. It is the lab mouse that is the exception. It remains an interesting historical artefact that the extreme susceptibility of the lab mouse led to ROP kinase discovery*.

A wild mouse homozygous for lab mouse chromosome 11 haplotypes would almost certainly be susceptible to virulent strains of *T. gondii* (though perhaps there are some resistant haplotypes on chromosome 18, even if not in CIM). Note that no other locus in the (CIM x BL/6) x BL/6 cross was able to compensate for the homozygous susceptible Chr 11 locus. Yet these mice are mongrels for the rest of the genome. We haven't found a perfect lab haplotype in a wild mouse yet although some are pretty close, but we haven't looked at very many yet either. Are the referees proposing that the haplotype itself is a laboratory artefact?

Presumably mice with these haplotypes are out there somewhere. We would rather say that the lab mouse *a* haplotypes, which are highly susceptible to virulent strains, constitute an accidental outcome of the breeding history of the mouse rather than an artefact (unless, as mentioned above, other haplotypes carry a cost under lab conditions).

*Along these lines, little is said about the coincidence between different IRG genotypes and* T. gondii *strains in the wild, which is obviously important to determine the significance of the laboratory observations*.

To our knowledge, nothing is known about this. The relevance and importance of the IRG system in controlling *T. gondii* is a recent discovery; its genetics is the content of this paper and has not been reported before. The referees raise what is obviously an important development from the present study, a development in which the present authors are presently fully engaged.